# Utility of polygenic embryo screening for disease depends on the selection strategy

Todd Lencz[1,2,3†]*, Daniel Backenroth[4†], Einat Granot-Hershkovitz[4], Adam Green[4], Kyle Gettler[5], Judy H Cho[5,6,7], Omer Weissbrod[8], Or Zuk[9], Shai Carmi[4]*

[1]Departments of Psychiatry and Molecular Medicine, Zucker School of Medicine at Hofstra/Northwell, Hempstead, United States; [2]Department of Psychiatry, Division of Research, The Zucker Hillside Hospital Division of Northwell Health, Glen Oaks, United States; [3]Institute for Behavioral Science, The Feinstein Institutes for Medical Research, Manhasset, United States; [4]Braun School of Public Health and Community Medicine, The Hebrew University of Jerusalem, Jerusalem, Israel; [5]Department of Genetics and Genomic Sciences, Icahn School of Medicine at Mount Sinai, New York, United States; [6]The Charles Bronfman Institute for Personalized Medicine, Icahn School of Medicine at Mount Sinai, New York, United States; [7]Department of Medicine, Icahn School of Medicine at Mount Sinai, New York, United States; [8]Department of Epidemiology, Harvard T.H. Chan School of Public Health, Boston, United States; [9]Department of Statistics and Data Science, The Hebrew University of Jerusalem, Jerusalem, Israel

*For correspondence:
tlencz@northwell.edu (TL);
shai.carmi@mail.huji.ac.il (SC)

†These authors contributed equally to this work

**Abstract** Polygenic risk scores (PRSs) have been offered since 2019 to screen in vitro fertilization embryos for genetic liability to adult diseases, despite a lack of comprehensive modeling of expected outcomes. Here we predict, based on the liability threshold model, the expected reduction in complex disease risk following *polygenic embryo screening* for a single disease. A strong determinant of the potential utility of such screening is the *selection strategy*, a factor that has not been previously studied. When only embryos with a very high PRS are excluded, the achieved risk reduction is minimal. In contrast, selecting the embryo with the lowest PRS can lead to substantial relative risk reductions, given a sufficient number of viable embryos. We systematically examine the impact of several factors on the utility of screening, including: variance explained by the PRS, number of embryos, disease prevalence, parental PRSs, and parental disease status. We consider both relative and absolute risk reductions, as well as population-averaged and per-couple risk reductions, and also examine the risk of pleiotropic effects. Finally, we confirm our theoretical predictions by simulating 'virtual' couples and offspring based on real genomes from schizophrenia and Crohn's disease case-control studies. We discuss the assumptions and limitations of our model, as well as the potential emerging ethical concerns.

## Introduction

Polygenic risk scores (PRSs) have become increasingly well-powered, relying on findings from large-scale genome-wide association studies for numerous diseases (*Visscher et al., 2017*; *Wray et al., 2013*). Consequently, a growing body of research has examined the potential clinical utility of applying PRSs in the treatment of adult patients in order to identify those at heightened risk for common late-onset diseases such as coronary artery disease or breast cancer (*Britt et al., 2020*; *Khera et al., 2018*; *Torkamani et al., 2018*). Another potential application of PRSs is preimplantation screening of in vitro fertilization (IVF) embryos, or *polygenic embryo screening* (PES). Polygenic embryo screening has been offered since 2019 (*Treff et al., 2019a*), but has been the focus of comparatively little

empirical research, despite debate over ethical and social concerns surrounding the practice (*Anomaly, 2020*; *Lázaro-Muñoz et al., 2021*; *Munday and Savulescu, 2021*).

We have recently demonstrated that screening embryos on the basis of polygenic scores for quantitative traits (such as height or intelligence) has limited utility in most realistic scenarios (*Karavani et al., 2019*), and that the accuracy of the score is a more significant determinant of PES utility for quantitative traits compared with the number of available embryos. On the other hand, a series of four studies (*Lello et al., 2020*; *Treff et al., 2019a*; *Treff et al., 2020*; *Treff et al., 2019b*) conducted by a private company providing PES services has suggested that PES for dichotomous disease risk may have significant clinical utility. However, these studies examined a relatively limited range of scenarios, primarily focusing on distinctions between sibling pairs discordant for illness, and did not provide a comprehensive examination of various potential PES settings. Filling this gap is an urgent need, as understanding the statistical properties of PES forms a critical foundation to any ethical consideration (*Lázaro-Muñoz et al., 2021*).

Here, we use statistical modeling to examine the potential utility of PES for reducing disease risk, with an aim toward informing future ethical deliberations. We focus on screening for a single complex disease, and study a range of realistic scenarios, quantifying the role of parameters such as the variance explained by the score, the number of available embryos, and the disease prevalence. We show that a major determinant of the outcome of PES is the *selection strategy*, namely the way in which an embryo is selected for implantation given the distribution of PRSs across embryos. We also study the risk reduction *conditional* on parental PRSs or disease status, and consider the risk of developing diseases not screened. Finally, we validate some of our predictions based on real genomes of cases and controls for two common complex diseases.

## Results

### Model and selection strategies

For each analysis presented below, we assume that a couple has generated, by IVF, $n$ viable embryos such that each embryo, if implanted, would have led to a live birth. We focus on a single complex disease, and assume that the corresponding PRS has been computed for each embryo. Given the PRSs of the $n$ embryos, a single embryo is selected for implantation based on a *selection strategy*.

The first strategy we consider is aimed only at avoiding high-risk embryos, consistent with studies of the potential clinical utility of PRSs in adults (*Chatterjee et al., 2016*; *Dai et al., 2019*; *Gibson, 2019*; *Khera et al., 2018*; *Mars et al., 2020*; *Mavaddat et al., 2019*; *Torkamani et al., 2018*). For example, the first case report presented on PES described the identification and exclusion of embryos with extremely high (top 2-percentiles) PRS (*Treff et al., 2019a*). We term this strategy 'high-risk exclusion' (HRE: *Figure 1A*, upper panel). Under HRE, after high-risk embryos are set aside, an embryo is randomly selected for implantation among the remaining available embryos. (In the case that all embryos are high-risk, we assume a random embryo is selected among them.)

An alternative selection strategy is to use the embryo with the lowest PRS. Ranking and prioritizing embryos for implantation based on morphology is common in current IVF practice (*Bormann et al., 2020*; *Montag et al., 2013*; *Rhenman et al., 2015*). If ranking is instead based on a disease PRS, the embryo with the lowest PRS could be selected, without any recourse to high-risk PRS thresholds. Such an approach was suggested by another recent publication from the same company (based on a multi-disease index), but outcomes were only examined in the context of sibling pairs (*Treff et al., 2020*). We term the implantation of the embryo with the lowest PRS as '*lowest-risk-prioritization*' (LRP; *Figure 1A*, lower panel).

In the following, we describe the theoretical risk reduction that can be achieved under these selection strategies. Our statistical approach is based on the liability threshold model (LTM; *Falconer, 1967*). The LTM represents disease risk as a continuous liability, comprising genetic and environmental risk factors, under the assumption that individuals with liability exceeding a threshold are affected. The liability threshold model has been shown to be consistent with data from family-based transmission studies (*Wray and Goddard, 2010*) and GWAS data (*Visscher and Wray, 2015*). Consequently, we define the disease risk of a given embryo probabilistically, as the chance that, given its PRS, its liability will cross the threshold at any point after birth (*Figure 1B*).

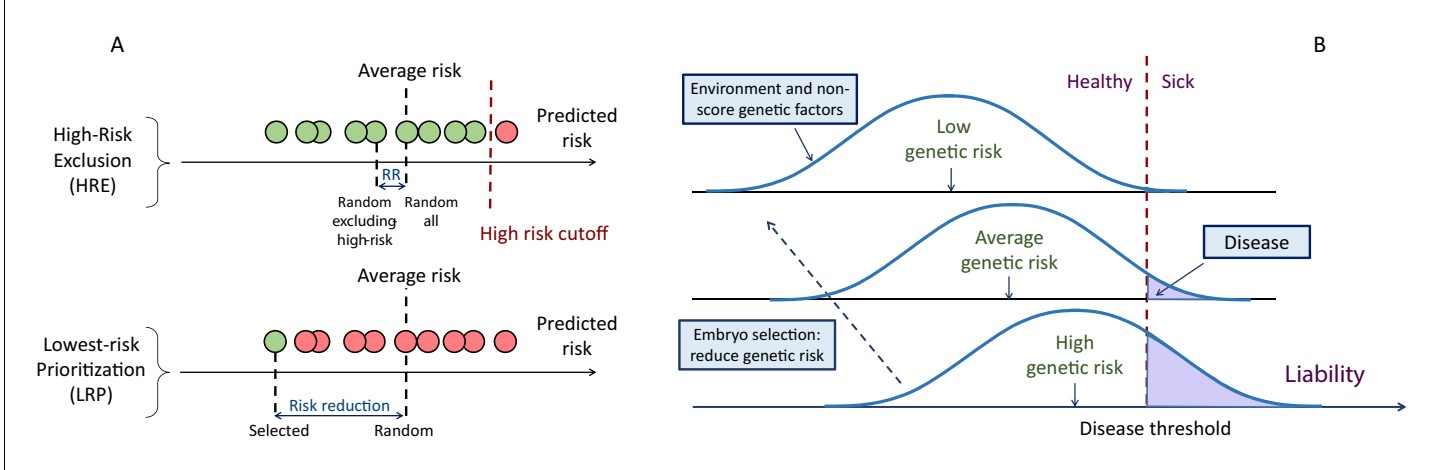

**Figure 1.** A schematic of the liability threshold model and polygenic embryo screening. (**A**) An illustration of the embryo selection strategies considered in this report. In the figure, each embryo is shown as a filled circle, and embryos are sorted based on their predicted risk, that is, their polygenic risk scores. Excluded embryos are shown in pink, and embryos that can be implanted in green. The risk reduction (RR) is indicated as the difference in risk between a randomly selected embryo (if no polygenic scoring was performed) and the embryo selected based on one of two strategies. In *high-risk exclusion* (HRE), the embryo selected for implantation is random, as long as its PRS is under a high-risk cutoff (usually the top few PRS percentiles). If all embryos are high-risk, a random embryo is selected. In *lowest-risk prioritization* (LRP), the embryo with the lowest PRS is selected for implantation. As we describe below, the LRP strategy yields much larger disease risk reductions. (**B**) An illustration of the liability threshold model (LTM). Under the LTM, each disease has an underlying (unobserved) liability, and an individual is affected if the total liability is above a threshold. The liability is composed of a genetic component and an environmental component, both assumed to be normally distributed in the population. For a given genetic risk (represented here by the polygenic risk score), the liability is the sum of that risk, plus a normally distributed *residual* component (environmental + genetic factors not captured by the PRS). For an individual with high genetic risk (bottom curve), even a modestly elevated (and thus, commonly-occurring) liability-increasing environment will lead to disease. For an individual with low genetic risk (top curve), only an extreme environment will push the liability beyond the disease threshold. Thus, disease risk reduction can be achieved with embryo screening by lowering the genetic risk of the implanted embryo. (Note that for the purpose of illustration, panel (**B**) displays three discrete levels of genetic risk, although in reality, the PRS is continuously distributed).

We use the following notation. We define the predictive power of a PRS as the proportion of variance in the liability of the disease explained by the score (*Dudbridge, 2013*), and denote it as $r_{\rm ps}^2$. We quantify the outcome of PES in two ways: the *relative risk reduction* (RRR) is defined as $\mathrm{RRR} = \frac{K - P(\text{disease})}{K} = 1 - \frac{P(\text{disease})}{K}$, where $K$ is the disease prevalence and $P(disease)$ is the probability of the selected embryo to be affected; the *absolute risk reduction* (ARR) is defined as $K - P(disease)$. For example, if a disease has prevalence of 5% and the selected embryo has a probability of 3% to be affected, the RRR is 40%, and the ARR is 2% points. We computed the RRR and ARR analytically under each selection strategy, and for various values for the disease prevalence, the strength of the PRS, embryo exclusion thresholds, and other parameters. The mathematical basis of the calculations is summarized in Materials and methods, and detailed in the Appendix.

## The risk reduction under the *high-risk exclusion* strategy

In *Figure 2* (upper row), we show the relative risk reduction achievable under the HRE strategy with $n = 5$ embryos. Under the 2-percentile threshold (straight black lines), the reduction in risk is limited: the RRR is <10% in all scenarios where $r_{\rm ps}^2 \leq 0.1$. Currently, $r_{\rm ps}^2 \approx 0.1$ (on the liability scale) is the upper limit of the predictive power of PRSs for most complex diseases (*Lambert et al., 2021*), with the exception of a few disorders with large-effect common variants (such as Alzheimer's disease or type 1 diabetes) (*Sharp et al., 2019*; *Zhang et al., 2020*). In the future, more accurate PRSs are expected. However, the common-variant SNP heritability is at most $\approx 30\%$ even for the most heritable diseases such as schizophrenia and celiac disease (*Holland et al., 2020*; *Zhang et al., 2018*), and it was recently suggested that $r_{\rm ps}^2 = 0.3$ is the maximal realistic value for the foreseeable future (*Wray et al., 2021*). At this value, relative risk reduction would be 20% for $K = 0.01$, 9% for $K = 0.05$, and 3% for $K = 0.2$. These gains achieved with HRE are small because the overwhelming

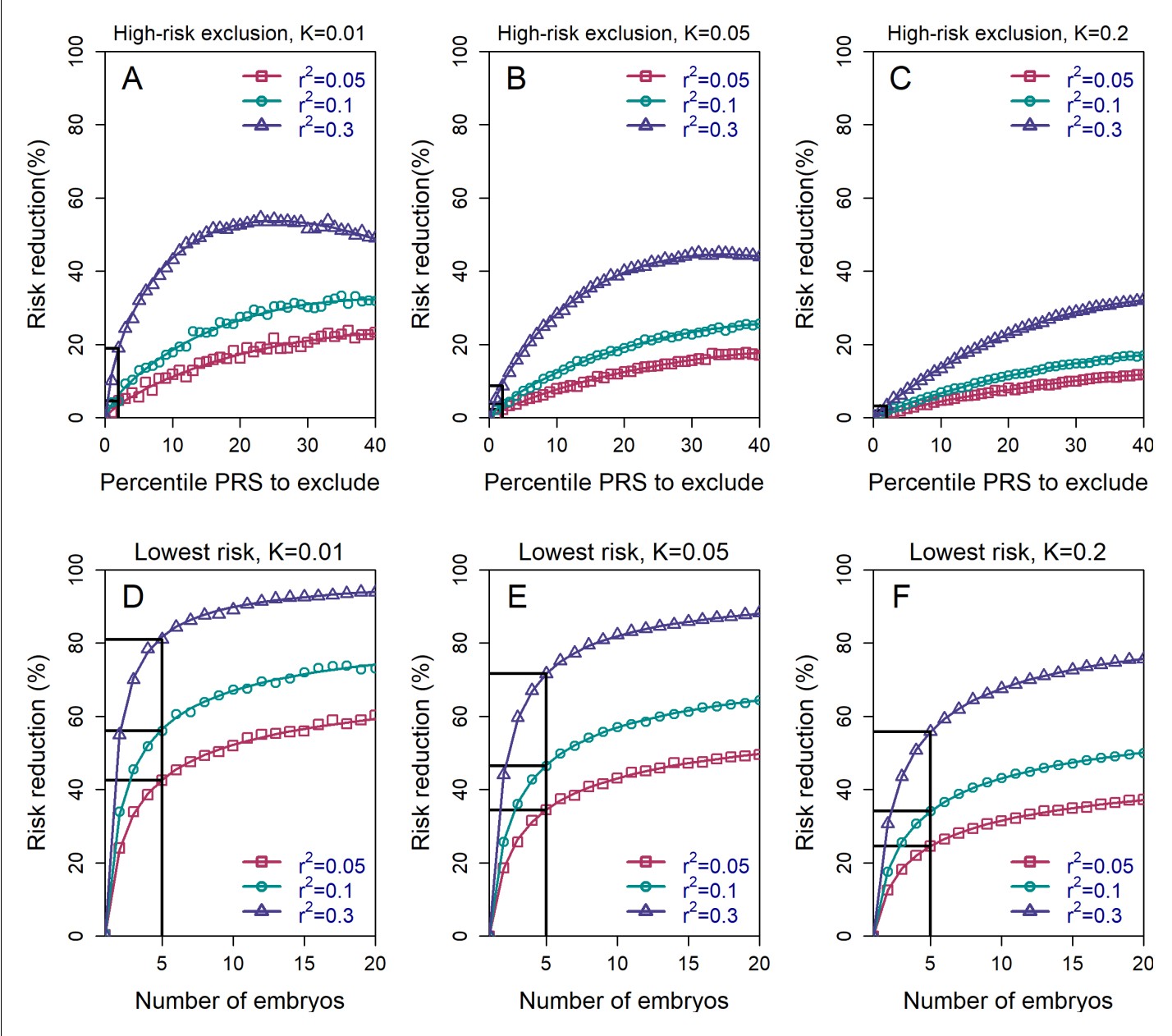

**Figure 2.** The relative risk reduction across selection strategies and disease parameters. The relative risk reduction (RRR) is defined as $(K − P(\text{disease}))/K$, where $K$ is the disease prevalence, and $P(\text{disease})$ is the probability of the implanted embryo to become affected. The RRR is shown for the high-risk exclusion (HRE) strategy in the upper row (panels (**A–C**)), and for the lowest-risk prioritization (LRP) in the lower row (panels (**D–F**)). See *Figure 1* for the definitions of the strategies. Results are shown for values of $K = 0.01, \; 0.05$ and $0.2$ (panels (**A–C**), respectively), and within each panel, for variance explained by the PRS (on the liability scale) $r_{ps}^2 = 0.05, 0.1$, and $0.3$ (legends). Symbols denote the results of Monte-Carlo simulations (Materials and methods), where PRSs of embryos were drawn based on a multivariate normal distribution, assuming PRSs are standardized to have zero mean and variance $r_{ps}^2$, and accounting for the genetic similarity between siblings (*Equation 4* in the Appendix). In each simulated set of $n$ sibling embryos ($n = 5$ for all simulations under HRE), one embryo was selected according to the selection strategy. The liability of the selected embryo was computed by adding a residual component (drawn from a normal distribution with zero mean and variance $1 − r_{ps}^2$) to its polygenic score. The embryo was considered affected if its liability exceeded $z_K$, the (upper) $K$-quantile of the standard normal distribution. We repeated the simulations over $10^6$ sets of embryos and computed the disease risk. In each panel, curves correspond to theory: *Equation (31)* in the Appendix for the HRE strategy, and *Equation (20)* in the Appendix for the LRP strategy. Black straight lines correspond to the RRR achieved when excluding embryos at the top 2% of the PRS (for HRE, upper panels) or for selecting the lowest risk embryo out of $n = 5$ (for LRP, lower panels).

The online version of this article includes the following figure supplement(s) for figure 2:

**Figure supplement 1.** The relative risk reduction for the high-risk exclusion strategy, with $n = 10$ available embryos.

*Figure 2 continued*

**Figure supplement 2.** The relative risk reduction under the high-risk exclusion (HRE) strategy, using two different rules for how an embryo is selected when all embryos are high risk.

**Figure supplement 3.** The relative risk reduction under the *lowest-risk prioritization* strategy for a dichotomized trait.

majority of affected individuals do not have extreme scores (*Murray et al., 2021*; *Wald and Old, 2019*).

Risk reduction increases as the threshold for exclusion is expanded to include the top quartile of scores, and then reaches a maximum at ≈25-50% under a range of prevalence and $r_{ps}^2$ values. For all these simulations, we set the number of available (testable) embryos to $n = 5$ (*Dahdouh, 2021*; *Sunkara et al., 2011*), although we acknowledge that the number of viable embryos may be much lower for many couples seeking IVF services for infertility (*Smith et al., 2015*). Simulations show that these estimates do not change much with increasing the number of embryos (see *Figure 2—figure supplement 1*). This holds especially at more extreme threshold values, since most batches of $n$ embryos will not contain any embryos with a PRS within, for example, the top 2-percentiles.

It should be noted that the relative risk reduction does not increase monotonically under HRE. Under our definition, whenever all embryos are high risk, an embryo is selected at random. Thus, at the extreme case when all embryos (i.e. top 100%) are designated as high risk, an embryo is selected at random at all times, and the relative risk reduction reduces to zero. We chose this definition of the HRE strategy because it does not involve ranking of the embryos. However, we can also consider an alternative strategy: if all embryos are high risk, the embryo with the lowest PRS is selected. Here, the RRR is expected to increase when increasing the threshold and designating more embryos as high risk, which we confirm in *Figure 2—figure supplement 2*. When the threshold is at 100% (all embryos are high risk), this alternative strategy (which we do not further consider) reduces to the *lowest-risk prioritization* strategy, which we study next.

## The risk reduction under the *lowest-risk prioritization* strategy

The HRE strategy treats all *non*-high-risk embryos equally. In practice, we expect most, or even all, embryos to be designated as non-high-risk, given the recent focus on the top PRS percentiles in the literature (e.g. *Khera et al., 2018*). However, as we have seen, this strategy leads to very little risk reduction. In *Figure 2* (lower panels), we show the expected RRR for the *lowest-risk prioritization* strategy, under which we prioritize for implantation the embryo with the lowest PRS, regardless of any PRS cutoff. Indeed, under the LRP strategy, risk reductions are substantially greater than in HRE. For example, with $n = 5$ available embryos, RRR>20% across the entire range of prevalence and $r_{ps}^2$ parameters considered, and can reach ≈50% for $K \leq 5\%$ and $r_{ps}^2 = 0.1$, and even ≈80% for $K = 1\%$ and $r_{ps}^2 = 0.3$. While RRR continues to increase as the number of available embryos increases, the gains are quickly diminishing after $n = 5$. On the other hand, *Figure 2* also demonstrates that RRR drops steeply if the number of embryos falls below $n = 5$, although the lower bound for RRR when just two embryos are available (≈20% for many scenarios) is still comparable to the upper bound of the HRE strategy for a greater number of embryos.

## Effects of PES on dichotomous vs quantitative traits

Our results demonstrate that, contrary to our previous study reporting only small effects of PES for quantitative traits (*Karavani et al., 2019*), PES can generate substantial relative risk reductions for diseases under the LRP strategy. To understand the relation between continuous and binary traits, consider an example involving IQ. Our estimate for the mean gain in IQ that could be achieved by selecting the embryo with the highest IQ polygenic score is approximately ≈2.5 IQ points (*Karavani et al., 2019*). Now assume that individuals with IQ<70 (2 SDs below the mean) are considered 'affected' according to a dichotomized trait of 'cognitive impairment'. Among individuals with IQ<70, the proportion of individuals with IQ in the range [67.5,70] is 33.5% (assuming a normal distribution). A gain of 2.5 points would shift such offspring beyond the threshold for 'cognitive impairment', resulting in a corresponding 33.5% reduction in risk of being 'affected'. (Note that the above explanation is intended to provide an intuition and ignores any variability in the gain.) *Figure 2—figure supplement 3* utilizes statistical modeling (with $r_{ps}^2$ derived from recent GWAS for

intelligence [*Savage et al., 2018*]) to demonstrate that substantial risk reductions can be achieved for a dichotomized trait, including when selecting out of just three embryos (panel (A)). Panel (B) extends these results to data for LDL cholesterol (with $r^2_{ps}$ derived from *Weissbrod et al., 2021*); given $n = 5$ embryos and the currently available PRS for LDL-C levels, risk reductions for 'high cholesterol' range from 40 to 60%, depending on the LDL level used to define the categorical trait. Thus, while implanting the embryo with the most favorable PRS is expected to result in very modest gains in an underlying quantitative trait, it is at the same time effective in avoiding embryos at the unfavorable tail of the trait.

## Effects of parental PRS and disease status

We next examined the effects of parental PRSs on the achievable risk reduction (Materials and methods, see also the Appendix), given that families with high genetic risk for a given disease may be more likely to seek PES. *Figure 3* demonstrates that, as expected, the HRE strategy shows greater relative risk reduction as parental PRS increases, in particular when excluding only very high-scoring embryos. This result follows directly from the fact that, on average, offspring will tend to have PRS scores near the mid-parental PRS value. In contrast, the relative RR (although not the *absolute* RR; see next section) for the LRP strategy somewhat declines as parental PRSs increase. Nevertheless, the RRR for the LRP strategy remains greater than that for the HRE strategy across all parameters (as expected by the definitions of these strategies).

It is also conceivable that families may be more likely to seek PES when one or both prospective parents is affected by a given disease. In *Figure 3—figure supplement 1*, we plot the RRR under the HRE and LRP strategies given that the parents are both healthy, both affected, or one of each (where we fixed the prevalence $K = 5\%$ and the heritability to $h^2 = 40\%$). The figure illustrates that parental disease status has relatively little impact on the expected RRR (especially in comparison to the changes under HRE when conditioning on the actual parental PRSs). This is because, as long as $r^2_{ps} \ll 1$, parental disease does not necessarily provide much information about parental PRS, and thus does not strongly constrain the number of risk alleles available to each embryo.

## Absolute vs relative risk

The above results were presented in terms of *relative* risk reductions. However, *Figure 3—figure supplement 1* also shows the baseline risk of an embryo of parents with a given disease status. For example, when one of the parents is affected, selecting the lowest risk embryo out of $n = 5$ (for a realistic $r^2_{ps} = 0.1$) reduces the risk from 10.0% to only 5.8%, thus nearly restoring the risk of the future child to the population prevalence (5%). More generally, we plot the *absolute* risk reduction (ARR) under the HRE and LRP strategies in *Figure 3—figure supplement 2* for a few values of the parental PRSs. Notably, while RRRs under the LRP strategy somewhat decrease with increasing parental PRSs, the ARRs substantially increase, in accordance with an expectation that PES in higher risk parents should eliminate more disease cases.

The clinical interpretation of these *absolute risk* changes will vary based on the population prevalence of the disorder (or the baseline risk of specific parents), and can offer a very different perspective on the magnitude of the effects (*Gordis, 2014*; *Lázaro-Muñoz et al., 2021*; *Murray et al., 2021*). In particular, for a *rare* disease, large *relative* risk reductions may result in very small changes in *absolute* risk. As an example, schizophrenia is a highly heritable (*Sullivan et al., 2003*) serious mental illness with prevalence of at most 1% (*Perälä et al., 2007*). The most recent large-scale GWAS meta-analysis for schizophrenia (*Ripke et al., 2020*) has reported that a PRS accounts for approximately 8% of the variance on the liability scale. Our model shows that a 52% RRR is attainable using the LRP strategy with $n = 5$ embryos. However, this translates to only $\approx 0.5$ percentage points reduction on the absolute scale: a randomly-selected embryo would have a 99% chance of not developing schizophrenia, compared to a 99.5% chance for an embryo selected according to LRP. In the case of a more common disease such as type 2 diabetes, with a lifetime prevalence in excess of 20% in the United States (*Geiss et al., 2014*), the RRR with $n = 5$ embryos (if the full SNP heritability of 17% [*Zhang et al., 2018*] were achieved) is 43%, which would correspond to >8 percentage points reduction in absolute risk.

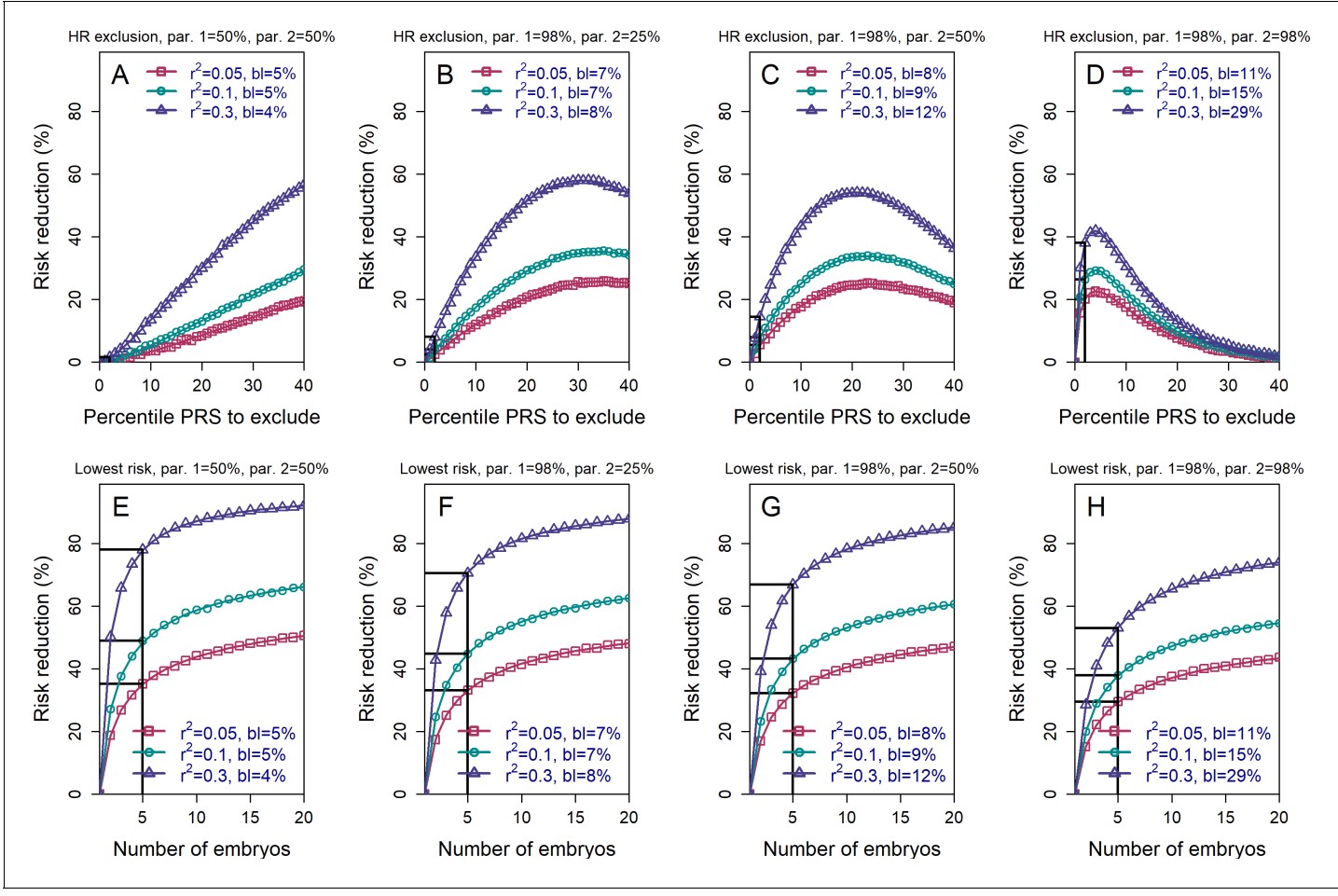

**Figure 3.** The relative risk reduction when the polygenic risk scores of the parents are known. Panels (A)-(D) are for the *high-risk exclusion* (HRE) strategy, while panels (E)-(H) are for the *lowest-risk prioritization* (LRP) strategy. All details are as in *Figure 2*, except the following. First, we fixed the prevalence to $K = 5\%$. Second, in the simulations, we drew the PRS of each embryo as $s_i = x_i + c$ ($i = 1, \ldots, n$), where $x_i$ is an embryo-specific component (independent across embryos) and $c$ is the shared component, also representing the mean parental PRS (Materials and methods). This is so far as in *Figure 2*; however, here we assumed that $c$ is given, equal to the average PRSs of the two parents. In each panel, we consider a different pair of PRSs for the parents. For example, in panels (A) and (E), both parents ('par. 1' and 'par. 2') have PRS equal to the 50% percentile of the PRS distribution; in panels (B) and (F), one parent has PRS equal to the 98% percentile of the PRS distribution, while the other has PRS equal to the 25% percentile; and so on. Third, in the simulations, we computed the risk reduction (according to either strategy) relative to a baseline, obtained from the same sets of simulations, when we always selected the first embryo. The baseline risk is indicated in each legend as 'bl'. Note that the baseline risk depends on the variance explained by the PRS, because the parental PRSs are determined as percentiles of the population distribution of the score, which has variance $r_{ps}^2$. Finally, we computed the theoretical disease risk for the HRE strategy using *Equation (29)* from the Appendix, the disease risk for the LRP strategy using *Equation (23)*, and the relative risk reduction (shown in curves) for both strategies using *Equation (36)*.

The online version of this article includes the following figure supplement(s) for figure 3:

**Figure supplement 1.** The relative risk reduction when the parental disease status is known.

**Figure supplement 2.** The *absolute* risk reduction when the polygenic risk scores of the parents are known.

## Variability of the risk reduction across couples

The results depicted in *Figure 2* describe the *average* risk reduction across the population, whereas the results in *Figure 3* demonstrate results for specific combinations of parental risk scores. However, it remains unclear whether the large average risk reductions observed under the LRP strategy are driven by only a small proportion of couples. More generally, we would like to fully characterize the dependence of the risk reduction on parental PRSs, which could be of interest to physicians and couples in real-world settings.

To address these questions, we define a new risk reduction index, which we term the *per-couple* relative risk reduction, or pcRRR. Informally, the pcRRR is the relative risk reduction conditional of the PRSs of the couple. Mathematically, $\mathrm{pcRRR}(\text{couple}) = 1 - \frac{P_s(\text{disease}|\text{couple})}{P_r(\text{disease}|\text{couple})}$. Here, $P_s(\text{disease}|\text{couple})$ is the probability that the (PRS-based) selected embryo is affected given the PRSs of the couple, and $P_r(\text{disease}|\text{couple})$ is similarly defined for a randomly selected embryo. Conveniently, the pcRRR depends only on the average of the maternal and paternal PRSs, which we denote as $c$. We calculated $\mathrm{pcRRR}(c)$ analytically under the LRP strategy (the Appendix), as well as computed the distribution of $\mathrm{pcRRR}(c)$ across all couples in the population.

We show the distribution of $\mathrm{pcRRR}(c)$ in *Figure 4*, panels (A)-(C). The results demonstrate that the pcRRR is relatively narrowly distributed around its mean, for all values of the prevalence ($K$) considered. The distribution becomes somewhat wider (and left-tailed) for the most extreme $r_{\mathrm{ps}}^2$ (0.3). Thus, the population-averaged RRRs are not driven by a small proportion of the couples. In agreement, the pcRRR depends only weakly on the average parental PRS, as can be seen in panels (D)-(F).

We note that the *per-couple* relative risk reduction is also an average, over all possible batches of $n$ embryos of the couple. One may thus ask what is the distribution of possible RRRs across these batches. We provide a short discussion in the Appendix (Section 5.3).

## Pleiotropic effects of selection on genetically negatively correlated diseases

Polygenic risk scores are often correlated across diseases (*Watanabe et al., 2019*; *Zheng et al., 2017*). Therefore, selecting based on the PRS of one disease may increase or decrease risk for other diseases. While a full analysis of screening for multiple diseases is left for future work, our simulation framework allows us to investigate the potential harmful effects of prioritizing embryos for one disease, in case that disease is negatively correlated with another disease (see the Appendix). We considered genetic correlations between diseases taking the values $\rho = (-0.05, -0.1, -0.15, -0.2, -0.3)$. [The most negative correlation between two diseases reported in LDHub (https://ldsc.broadinstitute.org/ldhub/) is $-0.3$, occurring between ulcerative colitis and chronic kidney disease (*Zheng et al., 2017*).] In general, negative correlations between diseases are uncommon, and when they occur, typical correlations are about $-0.1$.

*Figure 5* shows the simulated risk reduction for the target disease and the risk increase for the correlated disease, across different values of $\rho$ and for three values of the prevalence $K$ (panels (A)-(C); assumed equal for the two diseases), all under the LRP strategy. In all panels, we used $r_{\mathrm{ps}}^2 = 0.1$ for both diseases. The relative risk reduction for the target disease is, as expected, always higher in absolute value than the risk increase of the correlated disease. For typical values of $\rho = -0.1$ and $n = 5$, the relative increase in risk of the correlated disease is relatively small, at $\approx 6\%$ for $K \le 0.05$ and $\approx 3.5\%$ for $K = 0.2$. However, for strong negative correlation ($\rho = -0.3$) the increase in risk can reach 22%, 16%, or 11% for $K = 0.01, 0.05$ and $0.2$, respectively. Thus, care must be taken in the unique setting when the target disease is *strongly* negatively correlated with another disease.

## Simulations based on real genomes from case-control studies

Our analysis so far has been limited to mathematical analysis and simulations based on a statistical model. In principle, it would be desirable to compare our predictions to results based on real data. However, clearly, no real genomic and phenotypic data exist that would correspond to our setting, nor could such data be ethically or practically generated. Thus, we resort to a 'hybrid' approach, in which we simulate the genomes of embryos based on real genomic data from case-control studies. This approach is similar to the one we have previously used for studying polygenic embryo screening for traits (*Karavani et al., 2019*).

Briefly, our approach is as follows. We consider separately two diseases with somewhat differing genetic architecture: schizophrenia, which is amongst the most polygenic complex diseases, with no common loci of high effect size, and Crohn's disease, which is estimated to be less polygenic, and has several common loci with much larger effects than those found in schizophrenia (*O'Connor et al., 2019*). For each disease, we used genomes of unrelated individuals drawn from case-control studies. For schizophrenia, we used $\approx 900$ cases and $\approx 1600$ controls of Ashkenazi Jewish ancestry, while for Crohn's, we used $\approx 150$ cases and $\approx 100$ controls of European ancestry. We then generated 'virtual couples' by randomly mating pairs of individuals, regardless of sex. For each

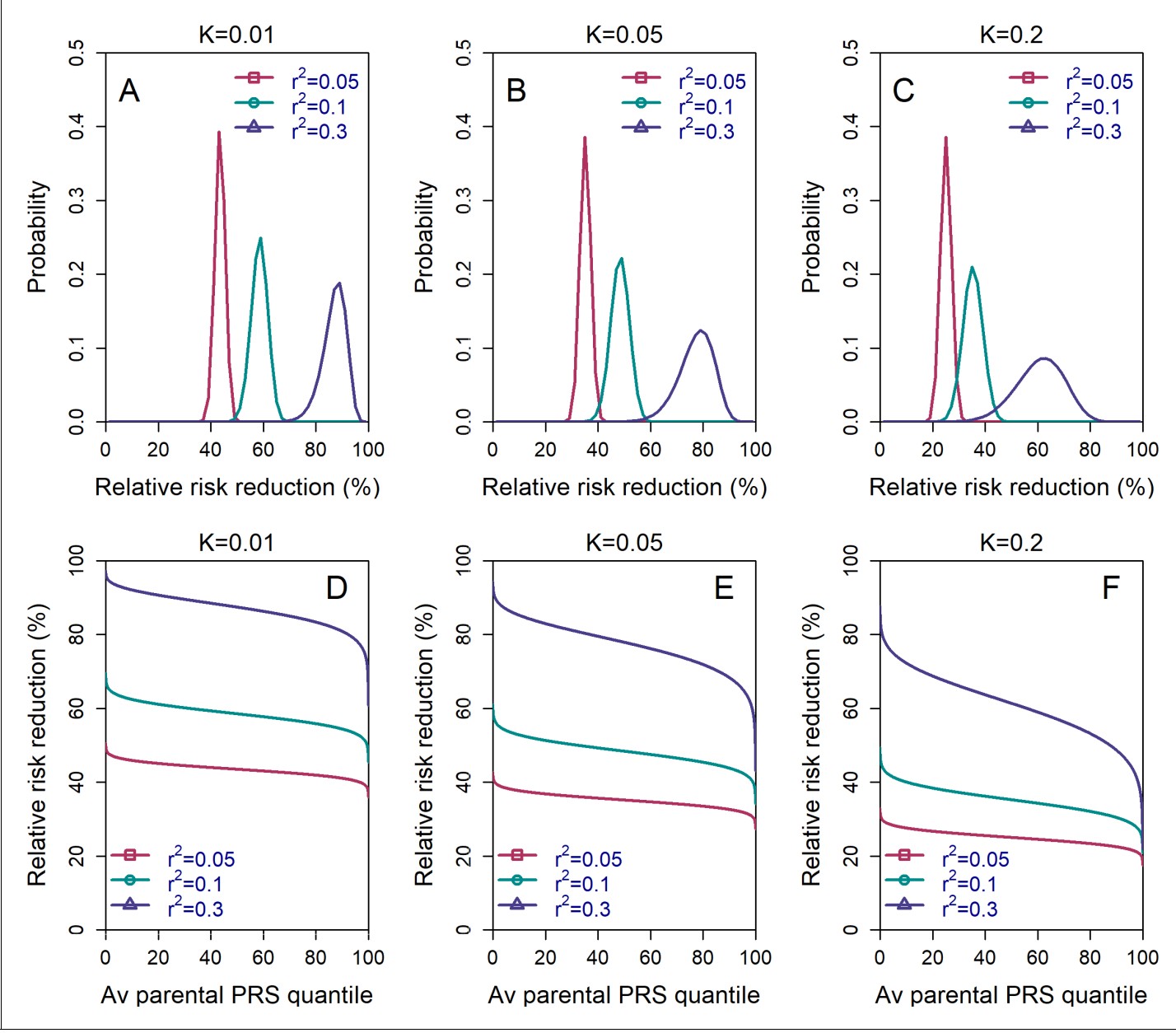

**Figure 4.** The variability in the relative risk reduction across couples. We considered only the *lowest-risk prioritization* strategy. In panels (**A–C**), we computed the theoretical distribution of the *per-couple* relative risk reduction, as explained in the Appendix Section 5. Briefly, the *per-couple* RRR is defined as $1 - P_s(\text{disease}|c)/P_r(\text{disease}|c)$, where $P_s(\text{disease}|c)$ is the probability of an embryo selected based on its PRS to be affected and $P_r(\text{disease}|c)$ is the probability of a randomly selected embryo to be affected, both conditional on the given couple. Our modeling suggests that $c$, which is the average of the paternal and maternal PRSs, is the only determinant of the relative risk reduction of a given couple. We computed the distribution of the *per-couple* RRR based on $10^4$ quantiles of $c$, thus covering all hypothetical couples in the population. The number of embryos was set to $n = 5$ in all panels. Panels (**A–C**) correspond to prevalence of $K = 0.01, 0.05$, and $0.02$, respectively. In panels (**D–F**), we plot the theoretical RRR vs the quantile of the average parental PRS $c$ (see Appendix Section 5.1).

couple, we simulate the genomes of $n$ hypothetical embryos, based on the laws of Mendelian inheritance and by randomly placing crossovers according to genetic map distances. In parallel, we used the 'parental' genomes to learn a logistic regression model that predicts the disease risk given a PRS computed based on existing summary statistics. We then computed the PRS of each simulated embryo, and predicted the risk that embryo to be affected. Finally, we compared the risk of disease

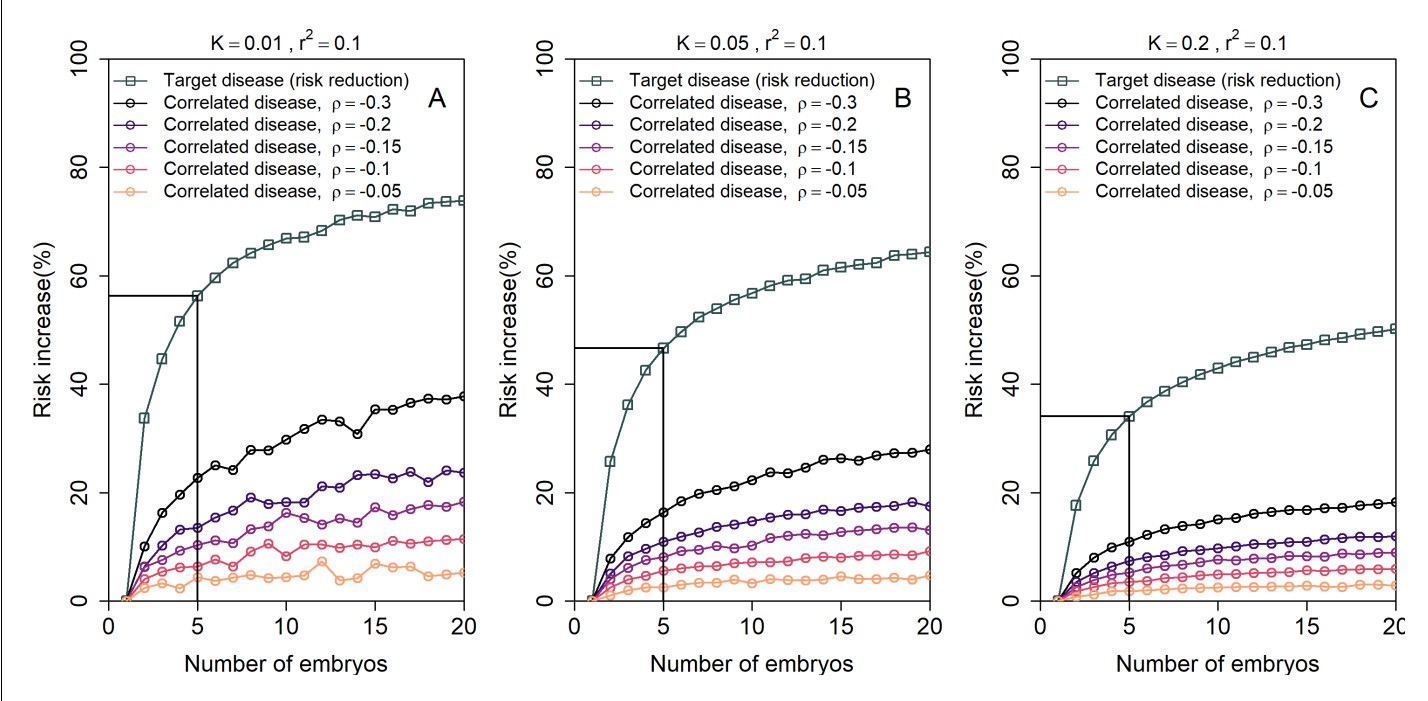

**Figure 5.** The increase in the risk of a negatively correlated disease due to polygenic embryo screening. We simulated two diseases that have genetic correlation $\rho<0$. We assumed that the prevalence $K$ is equal between the two diseases ($K = 0.01,\ 0.05$ and $0.2$: panels (**A**)-(**C**), respectively), and that $r_{\mathrm{ps}}^2 = 0.1$ for both diseases. We simulated polygenic scores for the two diseases in $n$ embryos in each of $10^6$ couples. For each couple, we selected the embryo either randomly or based on having the lowest PRS for the target disease. We then computed the risk of the embryo to have each disease as in the main analyses, by drawing the residual component of the liability and designating the embryo as affected if the total liability exceeded a threshold. The relative risk reduction of the target disease is shown as gray squares (and connecting lines) at the top of each plot. The relative risk *increase* for the correlated disease is shown in colored circles (and connecting lines), with different colors corresponding to different values of $\rho$ (see legend). Note that the risk reduction for the target disease is independent of $\rho$.

between a population in which one embryo per couple is selected at random, vs. a population in which one embryo is selected based on its PRS. For complete details, see Materials and methods.

In *Figure 6*, we plot the results for the relative risk reduction for schizophrenia (panels (A) and (B)) and Crohn's disease (panels (C) and (D)). For each disease, we consider both the HRE and LRP strategies. The analytical predictions closely match the empirical risk reductions generated in the simulations, except for a slight overestimation of the RRR under the LRP strategy. Nevertheless, for both schizophrenia and Crohn's disease, we empirically observe that RRRs as high as ≈45% are achievable with $n = 5$ embryos. In contrast, under the HRE strategy and when excluding embryos at the top 2% risk percentiles, risk reductions are very small, in agreement with the theoretical predictions. These results thus provide support to the robustness of our statistical model.

To further investigate the assumptions of our model, we test in *Figure 6—figure supplement 1* two intermediate predictions. The first is that the variance of the PRSs of embryos of a given couple should not depend on the average parental PRS. This is indeed the case (panels (A) and (C)), with the only exception of an uptick of the variance at very low parental PRSs for schizophrenia. The second prediction is that the variance across embryos is half of the variance in the parental population. The empirical results again show reasonable agreement with the theoretical prediction (panels (B) and (D)). The empirical variance (averaged across couples) was slightly lower than expected (by ≈4% for schizophrenia and ≈14% for Crohn's), which may explain our slight overestimation of the expected RRR under the LRP strategy.

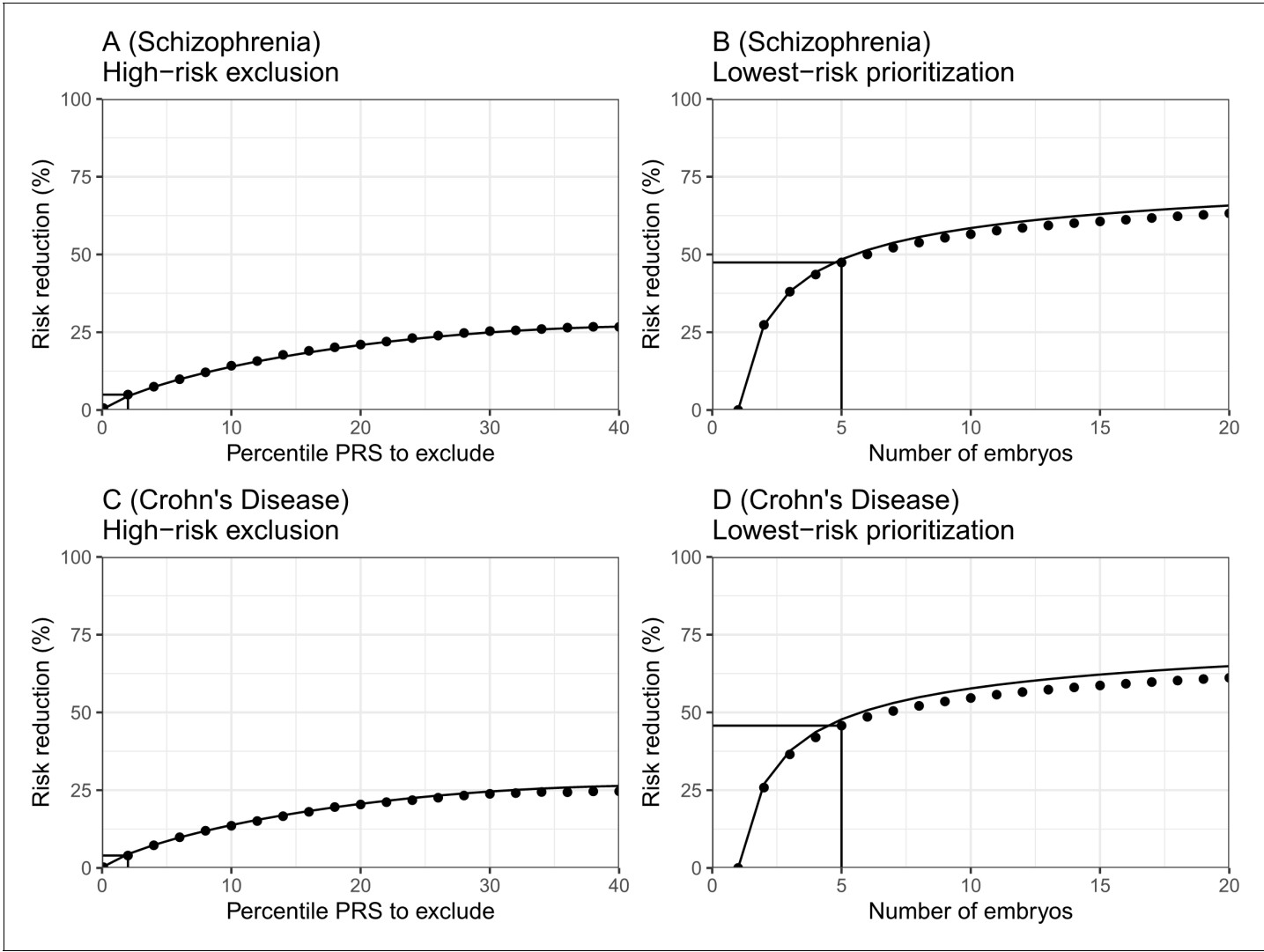

**Figure 6.** The empirical relative risk reduction in simulated embryos based on genomes from case-control studies of schizophrenia and Crohn's disease. We used $\approx 900$ cases and $\approx 1600$ controls for schizophrenia, and $\approx 150$ cases and $\approx 100$ controls for Crohn's. For each disease, we drew 5000 random 'virtual couples', regardless of sex, but correcting for case/control ascertainment. For each such random couple, we simulated the genomes of up to $n = 20$ embryos (children) based on Mendelian segregation and published recombination maps. For each embryo, we computed the PRS for the given disease (schizophrenia or Crohn's) using the most recent summary statistics that exclude our cohort. We computed the risk of each embryo to be affected based on a logistic regression model we learned in the 'parental' cohort. Panels (A) and (B) show results for schizophrenia, while panels (C) and (D) show results for Crohn's. In panels (A) and (C), we plot the relative risk reduction (RRR) under the *high-risk exclusion* (HRE) selection strategy, in which an embryo was randomly selected (out of $n = 5$ embryos), unless its PRS was above a given percentile. The RRR was computed against a baseline strategy of selection of an embryo at random and is plotted vs the exclusion percentile. In panels (B) and (D), we show the relative risk reduction under the *lowest-risk prioritization* (LRP) strategy, in which the embryo with the lowest PRS was selected. We plot the RRR vs the number of embryos $n$. In all panels, dots correspond to the results of simulations, and solid lines correspond to the theory. The theory was computed assuming prevalence of 1% for schizophrenia and 0.5% for Crohn's, and variance explained on the liability scale of $r_{ps}^2 = 0.068$ for schizophrenia $r_{ps}^2 = 0.056$ for Crohn's (calculated using the method of *Lee et al., 2012*). Further details are provided in Materials and methods.

The online version of this article includes the following figure supplement(s) for figure 6:

**Figure supplement 1.** The variance of the PRS across simulated embryos.

## Discussion

In this paper, we used statistical modeling to evaluate the expected outcomes of screening embryos based on polygenic risk scores for a single disease. We predicted the relative and absolute risk reductions, either at the population level or at the level of individual couples. Our model is flexible,

allowing us to provide predictions across various values of, for example, the PRS strength, the disease prevalence, the parental PRS or disease status, and the number of available embryos. We presented a comprehensive analysis of the expected outcomes across various settings, including when there is a concern about a second disease negatively correlated with the target disease. We finally validated our modeling assumptions using genomes from case-control studies. Our publicly available code could help researchers and other stakeholders estimate the expected outcomes for settings we did not cover.

Our most notable result was that a crucial determinant of risk reduction is the *selection strategy*. The use of PRS in adults has focused on those at highest risk (*Chatterjee et al., 2016*; *Dai et al., 2019*; *Gibson, 2019*; *Khera et al., 2018*; *Mars et al., 2020*; *Mavaddat et al., 2019*; *Torkamani et al., 2018*), for whom there may be maximal clinical benefit of screening and intervention. However, as PRSs have relatively low sensitivity, such a strategy is relatively ineffective in reducing the overall population disease burden (*Ala-Korpela and Holmes, 2020*; *Wald and Old, 2019*). Similarly, in the context of PES, exclusion of high-risk embryos will result in relatively modest risk reductions. By contrast, selecting the embryo with the lowest PRS may result in large reductions in relative risk.

While our prior work (*Karavani et al., 2019*) demonstrated that PES would have a small effect on quantitative traits, here we show that a small reduction in the liability can lead to a large reduction in the proportion of affected individuals. This is fundamentally a property of a threshold character with an underlying normally distributed continuous liability. For such traits, most of the individuals in the extreme of the liability distribution (i.e. the ones affected) are concentrated very near the threshold. Thus, even slightly reducing their liability can move a large proportion of affected individuals below the disease threshold. However, it should be noted that conventional thresholds for defining presence of disease may contain some degree of arbitrariness if the underlying distribution of pathophysiology is truly continuous. Consequently, the effects on ultimate morbidity may depend on the validity of the threshold itself (*Davidson and Kahn, 2016*).

We investigated how the range of potential PES outcomes varies with the PRSs of the parents or with their disease status. Under the HRE strategy, if only excluding embryos at the few topmost risk percentiles, the RRR is very small when the parents have low PRSs, and vice versa (*Figure 3*, panels (A)-(D)). This is expected, as excluding high PRS embryos will be effective only for couples who are likely to have many such embryos. Under the LRP strategy, the RRR depends only weakly on the parental PRSs (*Figure 3*, panels (E)-(H), and *Figure 4*). Under both strategies, the relative risk reduction depends only weakly on the parental disease status, as parental disease status is a weak signal for the underlying PRS. However, the *absolute* risk reduction increases substantially with increasing parental PRSs (*Figure 3—figure supplement 2*) and when one or more parents are affected.

Our study has several limitations. First, our results assume an infinitesimal genetic architecture for the disease, which may not be appropriate for oligogenic diseases and is not relevant for monogenic disorders. However, it has been repeatedly demonstrated that common, complex traits and diseases are highly polygenic (*Gazal et al., 2017*; *Holland et al., 2020*; *O'Connor et al., 2019*; *Shi et al., 2016*; *Zeng et al., 2018*; *Zeng et al., 2021*). For example, it was recently estimated that for almost all traits and diseases examined, the number of independently associated loci was at least $\approx 350$, reaching $\approx 10,000$ or more for cognitive and psychiatric phenotypes (*O'Connor et al., 2019*). This provides more than sufficient variability for the PRS to attain a normal distribution in the population and for our modeling assumptions to hold. Indeed, our empirical results for schizophrenia and Crohn's disease, two diseases with somewhat different genetic architectures, agreed reasonably well with the theoretical predictions. However, our models would need to be substantially adjusted in the presence of variants of very large effect, such as inherited or de novo coding variants or copy number variants, for example, as in autism (*Satterstrom et al., 2020*; *Takumi and Tamada, 2018*).

Additionally, our model relies on several simplifying statistical assumptions. For example, we did not explicitly model assortative mating, although this seems reasonable given that for genetic disease risk, correlation between parents is weak (*Rawlik et al., 2019*), and given that our previous study of traits showed no difference in the results between real and random couples (*Karavani et al., 2019*). This deficiency is also partly ameliorated by our modeling of the risk reduction when explicitly given the parental PRSs or disease status. Another assumption we made is that environmental influences on the child's phenotype are independent of those that have influenced the parents (when conditioning on the parental disease status). However, this is reasonable given

that family-specific environmental effect have been shown to be weak for complex diseases (*Wang et al., 2017*). For a discussion of additional model assumptions, see Appendix section 10.

Perhaps more importantly, we assumed throughout that $r_{\mathrm{ps}}^2$ represents the realistic accuracy of the PRS achievable, within-family, in a real-world setting in the target population. However, the realistically achievable $r_{\mathrm{ps}}^2$ may be lower than reported in the original publications that have generated the scores. For example, the accuracy of PRSs is sub-optimal when applied in non-European populations and across different socio-economic groups (*Duncan et al., 2019*; *Mostafavi et al., 2020*). A PRS that was tested on adults may be less accurate in the next generation. Additionally, the variance explained by the score, as estimated in samples of unrelated individuals, is inflated due to population stratification, assortative mating, and indirect parental effects (*Kong et al., 2018*; *Young et al., 2019*; *Morris et al., 2020*; *Mostafavi et al., 2020*). The latter, also called 'genetic nurture', refers to trait-modifying environmental effects induced by the parents based on their genotypes. These effects do not contribute to prediction accuracy when comparing polygenic scores between siblings (as when screening IVF embryos), and thus, the variance explained by polygenic scores in this setting can be substantially reduced, in particular for cognitive and behavioral traits (*Howe et al., 2021*; *Selzam et al., 2019*). Our risk reduction estimates thus represent an upper bound relative to real-world scenarios. On the other hand, recent empirical work on within-family disease risk prediction showed that the reduction in accuracy is at most modest (*Lello et al., 2020*), and within-siblings-GWAS yielded similar results to unrelated-GWAS for most physiological traits (*Howe et al., 2021*). Additionally, accuracy in non-European populations is rapidly improving due to the establishment of national biobanks in non-European countries (*Koyama et al., 2020*; *Vujkovic et al., 2020*) and improvement in methods for transferring scores into non-European populations (*Amariuta et al., 2020*; *Cai et al., 2021*). Either way, the analytical results presented in this paper are formulated generally as a function of the achievable accuracy $r_{\mathrm{ps}}^2$, and as such, users can substitute values relevant to their specific target population and disease.

Another major limitation of this work is that we have only considered screening for a single disease. In reality, couples may seek to profile an embryo on the basis of multiple disease PRSs simultaneously, or based a global measure of lifespan or healthspan (*Sakaue et al., 2020*; *Timmers et al., 2020*; *Zenin et al., 2019*). This is likely to reduce the per-disease risk reduction, as we have previously observed for quantitative traits (*Karavani et al., 2019*), but will also likely be more cost effective (*Treff et al., 2020*; *Gwern, 2018*). PES for multiple diseases requires the formulation and analysis of new selection strategies and is substantially more mathematically complex; we therefore leave it for future studies.

As our approach was statistical in nature, it is important to place our results in the context of real-world clinical practice of assisted reproductive technology. The number of embryos utilized in the calculations in the present study refers to viable embryos that could lead to *live birth*, which can be substantially smaller than the raw number of fertilized oocytes or even the number of implantable embryos at day 5. This consideration is especially important given the steep drop in risk reduction when the number of available embryos drops below 5 (*Figure 2*). In fact, many IVF cycles do not achieve *any* live birth. Rates of live birth decline with maternal age, in particular after age 40 (*Smith et al., 2015*); for women of age >42, fewer than 4% of IVF cycles result in live births, making PES impractical. On the other hand, success rates will likely be higher for young prospective parents who seek PES to reduce disease risk but do not suffer from infertility. However, the prospect of elective IVF for the purpose of PES in such couples must be weighed against the potential risks of these invasive procedures to the mother and child (*Dayan et al., 2019*; *Luke, 2017*).

A different concern is whether the embryo biopsy (which is required for genotyping) may cause risk to the viability and future health of the embryo. Several recent studies have demonstrated no evidence for potential adverse effects of trophectoderm biopsy on rates of successful implantation, fetal anomalies, and live birth (*Awadalla et al., 2021*; *He et al., 2019*; *Riestenberg et al., 2021*; *Tiegs et al., 2021*). Moreover, no significant adverse effects have been detected for postnatal child development in a recent meta-analysis (*Natsuaki and Dimler, 2018*). On the other hand, a number of studies have reported that trophectoderm biopsy was associated with pregnancy complications, including preterm birth, pre-eclampsia, and hypertensive disorders of pregnancy (*Li et al., 2021*; *Zhang et al., 2019*; *Makhijani et al., 2021*). Specific variations in biopsy protocols may account for differences in outcomes across studies (*Rubino et al., 2020*). Newly developed techniques may allow

in the future to genotype an embryo non-invasively based on DNA present in spent culture medium, although the accuracy of these methods is still being debated (*Leaver and Wells, 2020*). It should also be noted that, throughout this manuscript, we assumed the use of single embryo transfer.

Finally, the results of our study invite a debate regarding ethical and social implications. For example, the differential performance of PES across selection strategies and risk reduction metrics may be difficult to communicate to couples seeking assisted reproductive technologies (*Cunningham et al., 2015*; *Wilkinson et al., 2019*). Indeed, in the first PES case report, the couple elected to forego any implantation despite the availability of embryos that were designated as normal risk (*Treff et al., 2019a*). These difficulties are expected to exacerbate the already profound ethical issues raised by PES (as we have recently reviewed [*Lázaro-Muñoz et al., 2021*]), which include stigmatization (*McCabe and McCabe, 2011*), autonomy (including 'choice overload' [*Hadar and Sood, 2014*]), and equity (*Sueoka, 2016*). In addition, the ever-present specter of eugenics (*Lombardo, 2018*) may be especially salient in the context of the LRP strategy. How to juxtapose these difficulties with the potential public health benefits of PES is an open question. We thus call for urgent deliberations amongst key stakeholders (including researchers, clinicians, and patients) to address governance of PES and for the development of policy statements by professional societies. We hope that our statistical framework can provide an empirical foundation for these critical ethical and policy deliberations.

## Materials and methods

### Summary of the modeling results

In this section, we provide a brief overview of our model and derivations, with complete details appearing in the *Appendix*.

Our model is follows. We write the polygenic risk scores of a batch of $n$ IVF embryos as $(s_1, \ldots, s_n)$, and generate the scores as $s_i = x_i + c$. The $(x_1, \ldots, x_n)$ are embryo-specific independent random variables with distribution $x_i \sim N\left(0, r_{\mathrm{ps}}^2/2\right)$, $r_{\mathrm{ps}}^2$ is the proportion of variance in liability explained by the score, and $c$ is a shared component with distribution $c \sim N\left(0, r_{\mathrm{ps}}^2/2\right)$, also representing the average of the maternal and paternal scores.

In each batch, an embryo is selected according to the selection strategy. Under *high-risk exclusion*, we select a random embryo with score $s < z_q r_{\mathrm{ps}}$, where $z_q$ is the $(1-q)$-quantile of the standard normal distribution. If no such embryo exists, we select a random embryo, but we also studied the strategy when in such a case, the lowest scoring embryo is selected. Under *lowest-risk prioritization*, we select the embryo with the lowest value of $s$. We computed the liability of the selected embryo as $y = s + e$, where $e \sim N\left(0, 1 - r_{\mathrm{ps}}^2\right)$. We designate the embryo as affected if $y > z_K$, where $z_K$ is the $(1-K)$-quantile of the standard normal distribution and $K$ is the disease prevalence. In the simulations, we computed the disease probability (for each parameter setting) as the fraction of batches (out of $10^6$ repeats) in which the selected embryo was affected. We also simulated the score and disease status of a second disease, which is not used for selecting the embryo, but may be negatively correlated with the target disease.

We computed the disease probability analytically using the following approaches. We first computed the distribution of the score of the selected embryo. For *lowest-risk prioritization*, we used the theory of order statistics. For *high-risk exclusion*, we first conditioned on the shared component $c$, and then studied separately the case when all embryos are high-risk (i.e. have score $s > z_q r_{\mathrm{ps}}$), in which the distribution of the unique component of the selected embryo ($x$) is a normal variable truncated from below at $z_q r_{\mathrm{ps}} - c$, and the case when at least one embryo has score $s < z_q r_{\mathrm{ps}}$, in which $x$ is a normal variable truncated from above. We then integrated over the non-score liability components (and over $c$ in some of the settings) in order to obtain the probability of being affected. We solved the integrals in the final expressions numerically in R.

We computed the risk reduction based on the ratio between the risk of a child of a random couple when the embryo was selected by PRS and the population prevalence. We also provide explicit results for the case when the average parental PRS $c$ is known. These expressions allowed us to compute the distribution of risk reductions *per-couple*. Finally, when conditioning on the parental

disease status, we integrated the disease probability of the selected embryo over the posterior distribution of the parental score and non-score genetic components. For full details and for an additional discussion of previous work and limitations, see the *Appendix*. R code is available at: https://github.com/scarmi/embryo_selection (copy archived at swh:1:rev:4cdc572582deb9b745e6844d96e0344914f4595e, *Carmi, 2021*) and https://github.com/dbackenroth/embryo_selection (copy archived at swh:1:rev:c65bf082fcb28434c271260560c4a4450dad76a3,; *Backenroth, 2021*).

## Simulations based on genomes from case-control studies

Our main analysis has been limited to mathematical modeling of polygenic scores and their relation to disease risk. For obvious ethical and practical reasons, we could not validate our modeling predictions with actual experiments. Nevertheless, we could perform realistic simulations based on genomes from case-control studies, similarly to our previous work (*Karavani et al., 2019*). Our approach is as follows. We consider, separately, two diseases: schizophrenia and Crohn's. For schizophrenia, we use $\approx 900$ cases and $\approx 1600$ controls of Ashkenazi Jewish ancestry, while for Crohn's, we use $\approx 150$ cases and $\approx 100$ controls from the New York area. For each disease, we use these individuals, who are unrelated, to generate 'virtual couples' by randomly mating pairs of individuals. For each such 'couple', we simulate the genomes of $n$ hypothetical embryos, based on the laws of Mendelian inheritance and by randomly placing crossovers according to genetic map distances. In parallel, we use the same genomes to derive a logistic regression model that predicts the risk of disease given a PRS computed from the most recently available summary statistics (based on datasets not including the samples in our test cohorts). We then compute the PRS of each simulated embryo, and predict the risk of disease of that embryo. We finally compare the risk of disease between one randomly selected embryo per couple vs one embryo selected based on PRS. In the paragraphs below, we provide additional details.

### The Ashkenazi schizophrenia cohort

The samples and the genotyping process were previously described (*Lencz et al., 2013*). Patients were recruited from hospitalized inpatients at seven medical centres in Israel and were diagnosed with schizophrenia or schizoaffective disorder. Samples from healthy Ashkenazi individuals were collected from volunteers at the Israeli Blood Bank. All subjects provided written informed consent, and corresponding institutional review boards and the National Genetic Committee of the Israeli Ministry of Health approved the studies. DNA was extracted from whole blood and genotyped for ~1 million genome-wide SNPs using Illumina HumanOmni1-Quad arrays. We performed the following quality control steps. First, we removed samples with (1) genotyping call rate <95%; (2) one of each pair of related individuals (total shared identical-by-descent (IBD) segments >700cM); and (3) sharing of less than 15 cM on average with the rest of the cohort (indicating non-Ashkenazi ancestry). We removed SNPs with (1) call rate <97%; (2) minor allele frequency <1%; (3) significantly different allele frequencies between males and females (p-value threshold = 0.05/#SNPs); (4) differential missingness between males and females (p<$10^{-7}$) based on a $\chi^2$ test; (5) deviations from Hardy-Weinberg equilibrium in females (p-value threshold = 0.05/#SNPs); (6) SNPs in the HLA region (chr6:24–37M); and (7) (after phasing) SNPs having A/T or C/G polymorphism, as we could not unambiguously link them to corresponding effect sizes in the summary statistics. We finally used autosomal SNPs only. The remaining number of individuals was 2526 (897 cases and 1629 controls), and the number of SNPs was 728,505. We phased the genomes using SHAPEIT v2 (*Delaneau et al., 2013*).

### The Mt Sinai Crohn's disease cohort

Samples from subjects with Crohn's disease were recruited from clinics by Mt Sinai providers. All subjects provided written, informed consent in studies approved by the Mt Sinai Institutional Review Board. Genotyping was performed at the Broad Institute using the Illumina Global Screening Array (GSA) chip, as previously described (*Gettler et al., 2021*). We phased the genomes using Eagle v2.4.1 (*Loh et al., 2016*). We then removed SNPs having A/T or C/G polymorphism. The remaining number of individuals was 257 (154 cases and 103 controls) and the number of SNPs was 560,612.

## Simulating couples and embryos

For each disease, we generated 5000 unique couples by randomly pairing individuals (regardless of their sex) according to the population prevalence of the disease. For example, for schizophrenia, assuming a prevalence of 1%, a proportion $0.99^2$ of the couples were both controls. Given a pair of parents, we simulated 20 offspring (embryos) by specifying the locations of crossovers in each parent. Recombination was modeled as a Poisson process along the genome, with distances measured in cM using sex-averaged genetic maps (*Bhérer et al., 2017*). Specifically, for each parent and embryo, we drew the number of crossovers in each chromosome from a Poisson distribution with mean equal to the chromosome length in Morgan. We then determined the locations of the crossovers by randomly drawing positions along the chromosome (in Morgan). We mixed the phased paternal and maternal chromosomes of the parent according to the crossover locations, and randomly chose one of the resulting sequences as the chromosome to be transmitted to the embryo. We repeated for the other parent, in order to form the diploid genome of the embryo.

## Developing a polygenic risk score for schizophrenia

We used summary statistics from the most recent schizophrenia GWAS of the Psychiatric Genomics Consortium (PGC) (*Ripke et al., 2020*). Note that we specifically used summary statistics that excluded our Ashkenazi cohort. We used our entire cohort (2526 individuals) to estimate linkage disequilibrium (LD) between SNPs, and performed LD-clumping on the summary statistics in PLINK (*Chang et al., 2015*), with a window size of 250kb, a minimum $r^2$ threshold for clumping of 0.1, a minimum minor allele frequency threshold of 1%, and a maximum p-value threshold of 0.05. The p-value threshold was chosen based on results from the PGC study. After clumping, the final score included 23,036 SNPs. To construct the score, we used the effect sizes reported in the GWAS summary statistics, without additional processing.

## Developing a polygenic risk score for Crohn's disease

We used summary statistics derived from European samples available from https://www.ibdgenetics.org/downloads.html (*Liu et al., 2015*), which did not include our cohort. We estimated LD using the entire Crohn's disease cohort, and performed LD-clumping and p-value thresholding using the same parameters as for the schizophrenia cohort, as described above. The final score included 9,403 SNPs.

## Calculating the PRS and the risk of an embryo

For each disease, we calculated polygenic scores for each parent and simulated embryo in PLINK, using the `-score` command with default parameters. Using the polygenic scores of the parents, we fitted a logistic regression model for the case/control status as a function of the polygenic scores. We did not adjust for additional covariates: for schizophrenia, genetic ancestry is homogeneous in our Ashkenazi cohort, and age and sex contributed very little to predictive power (increased AUC from 0.695 only to 0.717). For Crohn's, age was not available, and sex did not contribute to predictive power (increased AUC from to 0.693 to 0.695). We adjusted the intercept of the logistic regression models to account for the case-control sampling (*Rose and van der Laan, 2008*). We then used the model to predict the probability that a simulated embryo would develop the disease.

To determine the percentiles of the PRS for each disease, we derived an approximation to the distribution of the PRS in the population by fitting a normal distribution to the scores in our dataset. To take into account the case/control ascertainment, we weighted the case and control samples according to the population prevalence of the disease (1% for schizophrenia [*Perälä et al., 2007*] and 0.5% for Crohn's [*GBD 2017 Inflammatory Bowel Disease Collaborators, 2020*]). We calculated the weighted mean and variance of the scores using the wtd.mean and wtd.var functions in the HMisc package in R. A normal distribution with the resulting mean and variance was used to calculate percentiles of the scores. The percentiles were then used to select (simulated) embryos under the *high-risk exclusion* strategy (see below).

## Calculating the risk reduction

For each disease, we performed the following simulations. For each selection strategy (either *high-risk exclusion* or *lowest-risk prioritization*), we selected one embryo for each couple according to the

strategy, and computed the probability of disease for the selected embryo. We then averaged the risk over all couples. We similarly computed the risk under selection of a random embryo for each couple. We computed the relative risk reduction based on the ratio between the risk under PRS-based selection and the risk under random selection. To compare to the theoretical expectations, we estimated the variance explained by the score on the liability scale using the method of *Lee et al., 2012*. Specifically, we first computed the correlation between the observed case/control status (coded as 1 and 0, respectively) and the PRS, and then used Equation (15) in Lee et al to convert the squared correlation to the variance explained. We obtained $r_{ps}^2 = 6.8\%$ for schizophrenia, which is close to the 7.7% reported in the original GWAS paper (*Ripke et al., 2020*), and $r_{ps}^2 = 5.6\%$ for Crohn's disease. We then substituted this value and prevalence of $K = 0.01$ for schizophrenia and $K = 0.005$ for Crohn's in our formulas for the relative risk reduction.

## Acknowledgements

We thank Gabriel Lázaro-Muñoz, Stacey Pereira, Chaim Jalas, and David A Zeevi for helpful discussions. This work was supported, in part, by a grant to Dr. Lencz and Dr. Carmi from the National Human Genome Research Institute (NHGRI) of the National Institutes of Health (NIH) under award number R01HG011711.

## Additional information

### Competing interests

Daniel Backenroth: Employee and shareholder at The Janssen Pharmaceutical Companies of Johnson & Johnson. Shai Carmi: Paid consultant to MyHeritage. The other authors declare that no competing interests exist.

### Funding

| Funder | Grant reference number | Author |
| --- | --- | --- |
| National Institutes of Health | R01HG011711 | Todd Lencz<br>Shai Carmi |

The funders had no role in study design, data collection and interpretation, or the decision to submit the work for publication.

### Author contributions

Todd Lencz, Conceptualization, Supervision, Investigation, Methodology, Funding Acquisition, Writing - original draft, Writing - review and editing; Daniel Backenroth, Formal analysis, Investigation, Methodology, Writing - review and editing, Software; Einat Granot-Hershkovitz, Formal analysis, Writing - review and editing; Adam Green, Formal analysis, Investigation; Kyle Gettler, Judy H Cho, Data curation, Writing - review and editing; Omer Weissbrod, Formal analysis, Investigation, Methodology, Writing - review and editing; Or Zuk, Conceptualization, Formal analysis, Supervision, Investigation, Methodology, Writing - review and editing; Shai Carmi, Conceptualization, Software, Formal analysis, Supervision, Funding acquisition, Investigation, Visualization, Methodology, Writing - original draft, Writing - review and editing

### Author ORCIDs

Todd Lencz https://orcid.org/0000-0001-8586-338X
Judy H Cho http://orcid.org/0000-0002-7959-0466
Shai Carmi https://orcid.org/0000-0002-0188-2610

### Decision letter and Author response

Decision letter https://doi.org/10.7554/eLife.64716.sa1
Author response https://doi.org/10.7554/eLife.64716.sa2

## Additional files

### Supplementary files

• Transparent reporting form

### Data availability

The modeling part of the study utilized simulated data. All code can be found at https://github.com/scarmi/embryo_selection (copy archived at https://archive.softwareheritage.org/swh:1:rev:4cdc572582deb9b745e6844d96e0344914f4595e) and https://github.com/dbackenroth/embryo_selection (copy archived at https://archive.softwareheritage.org/swh:1:rev:c65bf082fcb28434c271260560c4a4450dad76a3).

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

## Appendix 1

### 1 The liability threshold model

The liability threshold model (LTM) is a classic model in quantitative genetics (*Dempster and Lerner, 1950*; *Falconer, 1965*; *Lynch and Walsh, 1998*) and is also commonly used to analyze modern data (e.g. *Wray and Goddard, 2010*; *So et al., 2011*; *Lee et al., 2011*; *Lee et al., 2012*; *Do et al., 2012*; *Hayeck et al., 2017*; *Weissbrod et al., 2018*; *Hujoel et al., 2020*). Under the LTM, a disease has an underlying 'liability', which is normally distributed in the population, and is the sum of two components: genetic and non-genetic (the environment). Further, the LTM assumes an 'infinitesimal', or 'polygenic' genetic basis, under which a very large number of genetic variants of small effect combine to form the genetic component. An individual is affected if his/her total liability (genetic + environmental) exceeds a threshold.

Mathematically, if we denote the liability as $y$, the LTM can be written as

$$y = g + \epsilon, \tag{1}$$

where $y \sim N(0,1)$ is a standard normal variable, $g \sim N(0,h^2)$ is the genetic component, with variance equals to the heritability $h^2$, and $\epsilon \sim N(0, 1-h^2)$ is the non-genetic component. In practice, we cannot measure the genetic component, but only estimate it imprecisely with a polygenic risk score, denoted $s$. Following previous work (*So et al., 2011*; *Do et al., 2012*; *Lee et al., 2012*; *Treff et al., 2019a*; *Karavani et al., 2019*), we assume that the LTM can be written, similarly to *Equation (1)*, as

$$y = s + e, \tag{2}$$

where $y \sim N(0,1)$ as above, $s \sim N(0, r_{\mathrm{ps}}^2)$, where $r_{\mathrm{ps}}^2$ is the proportion of the variance in liability explained by the score, and $e \sim N(0, 1-r_{\mathrm{ps}}^2)$ is the residual of the regression of the liability on $s$ (and is uncorrelated with $s$), representing environmental effects *as well as* genetic factors not accounted for by the score.

An individual is affected whenever his/her liability exceeds a threshold. The threshold is selected such that the proportion of affected individuals is equal to the prevalence $K$, that is, it is equal to $z_K$, the $(1-K)$-quantile of a standard normal variable. Thus,

$$P(\mathrm{disease}) = P(y > z_K) = K. \tag{3}$$

The model is illustrated in *Figure 1B* of the main text.

### 2 A model for the scores of $n$ IVF embryos

Consider the polygenic risk scores (for a disease of interest) of $n$ IVF embryos of given parents. We assume no information is known about the parents, or, in other words, that the parents are randomly and independently drawn from the population. The scores of the embryos have a multivariate normal distribution,

$$\boldsymbol{s} = (s_1, \ldots, s_n) = \mathrm{MVN}(\boldsymbol{0}_n, \boldsymbol{\Sigma}), \tag{4}$$

where the means form a vector $\boldsymbol{0}_n$ of $n$ zeros, and the $n \times n$ covariance matrix is

$$\boldsymbol{\Sigma} = r_{\mathrm{ps}}^2 \begin{pmatrix} 1 & \frac{1}{2} & \cdots & \frac{1}{2} \\ \frac{1}{2} & 1 & \cdots & \frac{1}{2} \\ \cdots & \cdots & \cdots & \cdots \\ \frac{1}{2} & \frac{1}{2} & \cdots & 1 \end{pmatrix}. \tag{5}$$

The diagonal elements of the matrix are simply the variances of the individual scores of each embryo. The off-diagonal elements represent the covariance between the scores of the embryos, who are genetically siblings. Based on standard quantitative genetic theory (*Lynch and Walsh, 1998*) (see also our previous paper [*Karavani et al., 2019*]), the covariance between the scores of two siblings is $\mathrm{Cov}(s_i, s_j) = \frac{1}{2}\mathrm{Var}(s)$, and hence the off-diagonal elements follow. [The non-score components (the $e$ terms in *Equation (2)*) are also correlated. The correlation between the genetic

components of $e$ is modeled in Section 6. Modeling the correlation between the environmental components was unnecessary in this paper – see Section 10].

As we showed in our previous work (*Karavani et al., 2019*), the scores can be written as a sum of two independent multivariate normal variables, $s = x + c$, with

$$
\begin{aligned}
x &= (x_1, \ldots, x_n) \sim \text{MVN}\left(\mathbf{0}_n, \frac{r_{\text{ps}}^2}{2} I_n\right) \text{ and} \\
c &= (c_1, \ldots, c_n) \sim \text{MVN}\left(\mathbf{0}_n, \frac{r_{\text{ps}}^2}{2} J_n\right),
\end{aligned}
\tag{6}
$$

where $\mathbf{0}_n$ is a vector of zeros of length $n$, $I_n$ is the $n \times n$ identity matrix, and $J_n$ is the $n \times n$ matrix of all ones. The $x_i$'s and $c_i$'s have the same marginal distribution, namely normal with zero mean and variance $r_{\text{ps}}^2/2$ each. However, the $x_i$'s are independent, whereas $c$ has a constant covariance matrix, which means that the $c_i$'s are $n$ identical copies of the same random variable,

$$
c_1 \sim N\left(0, \frac{r_{\text{ps}}^2}{2}\right) \text{ and } c_2 = c_3 = \cdots = c_n = c_1 \equiv c.
\tag{7}
$$

Thus, for each embryo $i = 1, \ldots, n$,

$$
s_i = x_i + c.
\tag{8}
$$

## 2.1 An alternative interpretation: conditioning on the average parental scores

The decomposition of the score in *Equation (8)* can also be interpreted as conditioning on the average score of the parents. To see that, write the maternal score as $s_m$ and the paternal score as $s_f$. The variables $(s_i, s_m, s_f)$ have a multivariate normal distribution,

$$
(s_i, s_m, s_f) \sim \text{MVN}\left(\begin{pmatrix} 0 \\ 0 \\ 0 \end{pmatrix}, \begin{pmatrix} r_{\text{ps}}^2 & \frac{r_{\text{ps}}^2}{2} & \frac{r_{\text{ps}}^2}{2} \\ \frac{r_{\text{ps}}^2}{2} & r_{\text{ps}}^2 & 0 \\ \frac{r_{\text{ps}}^2}{2} & 0 & r_{\text{ps}}^2 \end{pmatrix}\right).
\tag{9}
$$

In the above equation, the variances of all scores are equal to $r_{\text{ps}}^2$. The covariance terms are $\text{Cov}(s_i, s_m) = \text{Cov}(s_i, s_f) = \frac{1}{2}\text{Var}(s) = \frac{r_{\text{ps}}^2}{2}$, as the relatedness between between parent and child is the same as for a pair of siblings. We assume no correlation between the scores of the parents (i.e. no assortative mating, see Section 10 for a discussion). We are now interested in the conditional density of $s_i$ given $s_m$ and $s_f$. Using standard results for multivariate normal distributions, the conditional density of $s_i$ is $N(\mu, \sigma^2)$, where,

$$
\begin{aligned}
\mu &= \Sigma_{12} \Sigma_{22}^{-1} \begin{pmatrix} s_m \\ s_f \end{pmatrix}, \\
\sigma^2 &= \Sigma_{11} - \Sigma_{12} \Sigma_{22}^{-1} \Sigma_{21},
\end{aligned}
\tag{10}
$$

and

$$
\Sigma_{11} = r_{\text{ps}}^2, \Sigma_{12} = \begin{pmatrix} \frac{r_{\text{ps}}^2}{2} & \frac{r_{\text{ps}}^2}{2} \end{pmatrix}, \Sigma_{21} = \begin{pmatrix} \frac{r_{\text{ps}}^2}{2} \\ \frac{r_{\text{ps}}^2}{2} \end{pmatrix}, \Sigma_{22} = \begin{pmatrix} r_{\text{ps}}^2 & 0 \\ 0 & r_{\text{ps}}^2 \end{pmatrix}.
\tag{11}
$$

These matrices are the blocks forming the covariance matrix in *Equation (9)*. Carrying out the matrix calculations, we obtain

$$\mu \quad = \frac{s_m + s_f}{2},$$
$$\sigma^2 \quad = \frac{r_{\text{ps}}^2}{2}. \tag{12}$$

Thus,

$$s_i \sim N\left(\frac{s_m + s_f}{2}, \frac{r_{\text{ps}}^2}{2}\right) \equiv N(c, r_{\text{ps}}^2/2), \tag{13}$$

where we defined the shared component $c \equiv \frac{s_m+s_f}{2}$ as the average parental score. The variance of $c$ itself, across the population, is $\text{Var}\left(\frac{s_m+s_f}{2}\right) = \frac{2\text{Var}(s)}{4} = r_{\text{ps}}^2/2$. Thus, $c \sim N(0, r_{\text{ps}}^2/2)$. In a given family, $c$ is the same across all embryos. Thus, *Equation (13)* is equivalent to $s_i = c + x_i$, with $c \sim N(0, r_{\text{ps}}^2/2)$ and $x_i \sim N(0, r_{\text{ps}}^2/2)$ being an embryo-specific component.

An analogous result holds for the total genetic component of the embryo, $g_i$, simply by replacing the proportion of variance explained by the score ($r_{\text{ps}}^2$) with the heritability ($h^2$). In other words, if $g_m$ and $g_f$ are the maternal and paternal genetic components, respectively, then

$$g_i \sim N\left(\frac{g_m + g_f}{2}, \frac{h^2}{2}\right). \tag{14}$$

## 3 The disease risk when implanting the embryo with the lowest risk

We assume next that we select for implantation the embryo with the lowest polygenic risk score for the disease of interest. Our goal will be to calculate the probability of that embryo to be affected. Since $s_i = x_i + c$, the score of the selected embryo satisfies

$$s_{\min} \quad = \min(x_1 + c, \ldots, x_n + c)$$
$$= \min(x_1, \ldots, x_n) + c \tag{15}$$
$$= x_{\min} + c,$$

where we defined $x_{\min} = \min(x_1, \ldots, x_n)$. Denote by $i^*$ the index of the selected embryo ($x_{i^*} = x_{\min}$). The liability of the embryo with the lowest risk is thus

$$y_{i^*} \quad = s_{\min} + e_{i^*}$$
$$= x_{\min} + c + e_{i^*} \tag{16}$$
$$= x_{\min} + \tilde{e},$$

where $e_i$ is the non-score component of embryo $i$, and $\tilde{e} = c + e_{i^*}$. We have,

$$\text{Var}(\tilde{e}) = \text{Var}(c) + \text{Var}(e_{i^*}) = \frac{r_{\text{ps}}^2}{2} + (1 - r_{\text{ps}}^2) = 1 - \frac{r_{\text{ps}}^2}{2}. \tag{17}$$

Therefore, the liability of the selected embryo can be written as a sum of two (independent) variables: $x_{min}$, which is the minimum of $n$ *independent* (zero mean) normal variables with variance $r_{\text{ps}}^2/2$ each; and $\tilde{e}$, which is a normal variable with (zero mean and) variance $1 - r_{\text{ps}}^2/2$.

The distribution of $x_{min}$ can be computed based on the theory of order statistics,

$$P(x_{\min} > t) = [P(x > t)]^n = \left[1 - \Phi\left(\frac{t}{r_{\text{ps}}/\sqrt{2}}\right)\right]^n. \tag{18}$$

In the above equation, the minimum of $n$ variables is greater than $t$ if and only if all variables are greater than $t$. The distribution of each $x$ is normal with zero mean and variance $r_{\text{ps}}^2/2$, and hence $P(x > t) = 1 - \Phi\left(\frac{t}{r_{\text{ps}}/\sqrt{2}}\right)$, where $\Phi(\cdot)$ is the cumulative probability distribution (CDF) of a standard normal variable.

We can now compute the probability of the selected embryo to be affected by demanding that the total liability is greater than the threshold $z_K$. Denote the probability of disease as $P_s(\text{disease})$ ($s$ stands for *selected*). Conditional on $\tilde{e}$,

$$\begin{aligned} P_s(\text{disease}\,|\,\tilde{e}) &= P(y_{i^*}>z_K\,|\,\tilde{e}) \\ &= P(x_{\min}+\tilde{e}>z_K) \\ &= P(x_{\min}>z_K-\tilde{e}) \\ &= \left[1-\Phi\left(\frac{z_K-\tilde{e}}{r_{\text{ps}}/\sqrt{2}}\right)\right]^n, \end{aligned} \tag{19}$$

where in the fourth line, we used *Equation (18)*. Next, denote by $f(\tilde{e})$ the density of $\tilde{e}$, and by $\phi(\cdot)$ the probability density function of a standard normal variable. Given that $\tilde{e}\sim N(0,1-r_{\text{ps}}^2/2)$,

$$\begin{aligned} P_s(\text{disease}) &= \int_{-\infty}^{\infty} P_s(\text{disease}\,|\,\tilde{e})f(\tilde{e})d\tilde{e} \\ &= \int_{-\infty}^{\infty}\left[1-\Phi\left(\frac{z_K-\tilde{e}}{r_{\text{ps}}/\sqrt{2}}\right)\right]^n \frac{1}{\sqrt{1-r_{\text{ps}}^2/2}}\phi\left(\frac{\tilde{e}}{\sqrt{1-r_{\text{ps}}^2/2}}\right)d\tilde{e} \\ &= \int_{-\infty}^{\infty}\left[1-\Phi\left(\frac{z_K-t\sqrt{1-r_{\text{ps}}^2/2}}{r_{\text{ps}}/\sqrt{2}}\right)\right]^n \phi(t)dt. \end{aligned} \tag{20}$$

In the third line, we changed variables: $t=\tilde{e}/\sqrt{1-r_{\text{ps}}^2/2}$. *Equation (20)* is our final expression for the probability of the embryo with the lowest score to be affected.

## 3.1 The risk reduction when conditioning on the mean parental score

Consider the case when $c$ is given, or, in other words, when we know the mean parental polygenic score. Let us compute the disease risk in such a case. We start from *Equation (16)*,

$$\begin{aligned} y_{i^*} &= s_{\min}+e_{i^*} \\ &= x_{\min}+c+e_{i^*}. \end{aligned} \tag{21}$$

Then,

$$\begin{aligned} P_s(\text{disease}\,|\,c,e_{i^*}) &= P(y_{i^*}>z_K\,|\,e_{i^*}) \\ &= P(x_{\min}+c+e_{i^*}>z_K) \\ &= P(x_{\min}>z_K-c-e_{i^*}) \\ &= \left[1-\Phi\left(\frac{z_K-c-e_{i^*}}{r_{\text{ps}}/\sqrt{2}}\right)\right]^n, \end{aligned} \tag{22}$$

where in the last line, we used *Equation (18)*.

Finally, with $f(e_{i^*})$ denoting the density of $e_{i^*}$, and recalling that $e_{i^*}\sim N\left(0,1-r_{\text{ps}}^2\right)$,

$$\begin{aligned} P_s(\text{disease}\,|\,c) &= \int_{-\infty}^{\infty} P_s(\text{disease}\,|\,c,e_{i^*})f(e_{i^*})de_{i^*} \\ &= \int_{-\infty}^{\infty}\left[1-\Phi\left(\frac{z_K-c-e_{i^*}}{r_{\text{ps}}/\sqrt{2}}\right)\right]^n \frac{1}{\sqrt{1-r_{\text{ps}}^2}}\phi\left(\frac{e_{i^*}}{\sqrt{1-r_{\text{ps}}^2}}\right)de_{i^*} \\ &= \int_{-\infty}^{\infty}\left[1-\Phi\left(\frac{z_K-c-t\sqrt{1-r_{\text{ps}}^2}}{r_{\text{ps}}/\sqrt{2}}\right)\right]^n \phi(t)dt, \end{aligned} \tag{23}$$

where in the last line, we changed variables, $t=e_{i^*}/\sqrt{1-r_{\text{ps}}^2}$. *Equation (23)* thus provides the probability of disease when we are given the mean parental score $c$.

## 4 The disease risk when excluding high-risk embryos

We now consider the selection strategy in which the implanted embryo is selected at random, as long as its risk score is not particularly high. Specifically, we assume that whenever possible, embryos at the top $q$ risk percentiles are excluded. When *all* embryos have high risk, we assume that a random embryo is selected. Let $z_q$ be the $(1-q)$-quantile of the standard normal distribution. The variance of the score is $r_{\text{ps}}^2$, and therefore, the score of the selected embryo must be lower than $z_q r_{\text{ps}}$.

To compute the disease risk in this case, we first condition on the shared, family-specific component $c$. We later integrate over $c$ to derive the risk across the population. Denote by $x_s$ the value of $x$ for the *selected* embryo, and for the moment, also condition on $x_s$. We have,

$$
\begin{aligned}
P_s(\text{disease}\,|\,x_s,c) &= P(y{>}z_K\,|\,c) \\
&= P(s{+}e{>}z_K\,|\,c) \\
&= P(x_s{+}c{+}e{>}z_K) \\
&= P(e{>}z_K-x_s-c) \\
&= 1-\Phi\left(\frac{z_K-x_s-c}{\sqrt{1-r_{\text{ps}}^2}}\right),
\end{aligned}
\tag{24}
$$

To obtain $P_s(\text{disease}\,|\,c)$, we need to integrate over $f(x_s)$, the density of $x_s$. In fact, $f(x_s)$ is a mixture of two distributions, depending on whether or not all embryos were high risk. Denote by $H$ the event that all embryos are high risk, and let us first compute the probability of $H$. Recall that given $c$, the scores of all embryos, $s_i = x_i + c$, are independent. The event $H$ is equivalent to the intersection of the *independent* events $\{s_i{>}z_q r_{\text{ps}}\}$ for $i = 1,\ldots,n$. Thus, recalling that $x_i \sim N(0, r_{\text{ps}}^2/2)$,

$$
\begin{aligned}
P(H) &= \prod_{i=1}^{n} P(s_i{>}z_q r_{\text{ps}}) \\
&= \prod_{i=1}^{n} P(x_i{+}c{>}z_q r_{\text{ps}}) \\
&= \prod_{i=1}^{n} P(x_i{>}z_q r_{\text{ps}}-c) \\
&= \left[1-\Phi\left(\frac{z_q r_{\text{ps}}-c}{r_{\text{ps}}/\sqrt{2}}\right)\right]^n.
\end{aligned}
\tag{25}
$$

Given $H$, we know that all scores were higher than the cutoff, i.e., that $x_i{>}z_q r_{\text{ps}}-c$ for all $i = 1,\ldots,n$. An embryo is then selected at random. Thus, $x_s$, the value of $x$ of the selected embryo, is a realization of a normal random variable truncated from below. Specifically, if $f_x(\cdot)$ is the unconditional density of $x$, then for $x_s{>}z_q r_{\text{ps}}-c$,

$$
f(x_s\,|\,H) = \frac{f_x(x_s)}{P(x{>}z_q r_{\text{ps}}-c)} = \frac{\frac{1}{r_{\text{ps}}/\sqrt{2}}\phi\left(\frac{x_s}{r_{\text{ps}}/\sqrt{2}}\right)}{1-\Phi\left(\frac{z_q r_{\text{ps}}-c}{r_{\text{ps}}/\sqrt{2}}\right)}.
\tag{26}
$$

In the case $H$ did not occur, we select an embryo at random among embryos with score $s_i{<}z_q r_{\text{ps}}$, that is, $x_i{<}z_q r_{\text{ps}}-c$. The density of $x_s$ is again, analogously to the above case, a realization of a normal random variable, but this time truncated from above. For $x_s{<}z_q r_{\text{ps}}-c$,

$$
f(x_s\,|\,\overline{H}) = \frac{f_x(x_s)}{P(x{<}z_q r_{\text{ps}}-c)} = \frac{\frac{1}{r_{\text{ps}}/\sqrt{2}}\phi\left(\frac{x_s}{r_{\text{ps}}/\sqrt{2}}\right)}{\Phi\left(\frac{z_q r_{\text{ps}}-c}{r_{\text{ps}}/\sqrt{2}}\right)}.
\tag{27}
$$

Using these results, we can write the density of $x_s$ when conditioning only on $c$,

$$f(x_s) = \begin{cases} f(x_s \mid H)P(H) + 0 \cdot P(\overline{H}) & \text{for } x_s > z_q r_{\text{ps}} - c \\ 0 \cdot P(H) + f(x_s \mid \overline{H})P(\overline{H}) & \text{for } x_s < z_q r_{\text{ps}} - c \end{cases}$$

$$= \begin{cases} \dfrac{\frac{1}{r_{\text{ps}}/\sqrt{2}}\phi\left(\frac{x_s}{r_{\text{ps}}/\sqrt{2}}\right)}{1 - \Phi\left(\frac{z_q r_{\text{ps}} - c}{r_{\text{ps}}/\sqrt{2}}\right)}\left[1 - \Phi\left(\frac{z_q r_{\text{ps}} - c}{r_{\text{ps}}/\sqrt{2}}\right)\right]^n & \text{for } x_s > z_q r_{\text{ps}} - c \\[4mm] \dfrac{\frac{1}{r_{\text{ps}}/\sqrt{2}}\phi\left(\frac{x_s}{r_{\text{ps}}/\sqrt{2}}\right)}{\Phi\left(\frac{z_q r_{\text{ps}} - c}{r_{\text{ps}}/\sqrt{2}}\right)}\left\{1 - \left[1 - \Phi\left(\frac{z_q r_{\text{ps}} - c}{r_{\text{ps}}/\sqrt{2}}\right)\right]^n\right\} & \text{for } x_s < z_q r_{\text{ps}} - c \end{cases} \tag{28}$$

We can now integrate over all $x_s$, still conditioning on $c$, and using *Equation (24)* and some algebra,

$$P_s(\text{disease} \mid c) = \int_{-\infty}^{\infty} f(x_s) P_s(\text{disease} \mid x_s, c)\, dx_s$$

$$= \int_{-\infty}^{z_q r_{\text{ps}} - c} \frac{\frac{1}{r_{\text{ps}}/\sqrt{2}}\phi\left(\frac{x_s}{r_{\text{ps}}/\sqrt{2}}\right)}{\Phi\left(\frac{z_q r_{\text{ps}} - c}{r_{\text{ps}}/\sqrt{2}}\right)}\left\{1 - \left[1 - \Phi\left(\frac{z_q r_{\text{ps}} - c}{r_{\text{ps}}/\sqrt{2}}\right)\right]^n\right\}\left[1 - \Phi\left(\frac{z_K - x_s - c}{\sqrt{1 - r_{\text{ps}}^2}}\right)\right] dx_s$$

$$+ \int_{z_q r_{\text{ps}} - c}^{\infty} \frac{\frac{1}{r_{\text{ps}}/\sqrt{2}}\phi\left(\frac{x_s}{r_{\text{ps}}/\sqrt{2}}\right)}{1 - \Phi\left(\frac{z_q r_{\text{ps}} - c}{r_{\text{ps}}/\sqrt{2}}\right)}\left[1 - \Phi\left(\frac{z_q r_{\text{ps}} - c}{r_{\text{ps}}/\sqrt{2}}\right)\right]^n\left[1 - \Phi\left(\frac{z_K - x_s - c}{\sqrt{1 - r_{\text{ps}}^2}}\right)\right] dx_s$$

$$= \int_{-\infty}^{\infty} \eta(t, \gamma(c))\xi(t, c)\, dt, \tag{29}$$

where we defined

$$\xi(t, c) = \phi(t)\left[1 - \Phi\left(\frac{z_K - t r_{\text{ps}}/\sqrt{2} - c}{\sqrt{1 - r_{\text{ps}}^2}}\right)\right],$$

$$\eta(t, \gamma) = \begin{cases} \frac{1 - [1 - \Phi(\gamma)]^n}{\Phi(\gamma)} & \text{for } t < \gamma, \\ [1 - \Phi(\gamma)]^{n-1} & \text{for } t > \gamma. \end{cases}, \text{ and} \tag{30}$$

$$\gamma(c) = \sqrt{2}z_q - \frac{c}{r_{\text{ps}}/\sqrt{2}}.$$

*Equation (29)* provides an expression for the probability of a disease given the mean parental score $c$.

Finally, we can integrate over all $c$ in order to obtain the probability of disease in the population. Recalling that $c \sim N(0, r_{\text{ps}}^2/2)$ and denoting its density as $f(c)$, and again after some algebra,

$$P_s(\text{disease}) = \int_{-\infty}^{\infty} P_s(\text{disease} \mid c) f(c)\, dc$$

$$= \int_{-\infty}^{\infty} \phi(u)\left[\int_{-\infty}^{\infty} \eta(t, \beta(u))\zeta(u, t)\, dt\right] du, \tag{31}$$

where we defined

$$\zeta(u, t) = \phi(t)\left[1 - \Phi\left(\frac{z_K - (u + t)r_{\text{ps}}/\sqrt{2}}{\sqrt{1 - r_{\text{ps}}^2}}\right)\right] \tag{32}$$

$$\beta(u) = \sqrt{2}z_q - u,$$

and $\eta(t, \cdot)$ was defined in *Equation (30)* above. *Equation (31)* is our final expression for the probability of an embryo to be affected after being selected randomly among non-high-risk embryos.

## 5 The relative risk reduction

We define the relative risk reduction (RRR) as follows. We are given the prevalence $K$ and the probability of the selected embryo to be affected $P_s(\text{disease})$ (averaged over the population). Then,

$$\mathrm{RRR} = \frac{K - P_s(\mathrm{disease})}{K} = 1 - \frac{P_s(\mathrm{disease})}{K}. \tag{33}$$

The absolute risk reduction (ARR) is similarly defined as $K - P_s(\mathrm{disease})$. For example, if a disease has prevalence of 5% and an embryo selected based on PRS has an average probability of 3% to be affected, the relative risk reduction is 40%, while the absolute risk reduction is 2% points.

To use *Equation (33)*, $P_s(\mathrm{disease})$ is given by *Equation (20)* for the *lowest-risk prioritization* strategy, and by *Equation (31)* for the *high-risk exclusion* strategy. We solve the integrals in these equations numerically in R using the function integrate (see Section 11).

## 5.1 The *per-couple* relative risk reduction

The RRR, as defined in *Equation (33)*, is the (complement of the) ratio between two *average* risks: the average risk of a random couple that would select an embryo based on its PRS, and the average risk of a random couple that would select an embryo at random. It can also be seen as the relative risk reduction between the risks in two hypothetical 'populations': one in which all embryos are selected based on a PRS-based strategy, and one in which all embryos are selected at random.

However, a shortcoming of the population-level RRR definition is that it does not provide information on the risk reduction expected for *individual couples*. In other words, a given couple may wish to know the extent to which they can reduce disease risk in their children by electing to select an embryo based on PRS. Conveniently, the only relevant information that characterizes the potential risk reduction for a given couple (in the absence of phenotypic data) is $c$, the average parental score.

We define the *per-couple* relative risk reduction, or $\mathrm{pcRRR}(c)$, as

$$\mathrm{pcRRR}(c) = \frac{P_r(\mathrm{disease}\,|\,c) - P_s(\mathrm{disease}\,|\,c)}{P_r(\mathrm{disease}\,|\,c)} = 1 - \frac{P_s(\mathrm{disease}\,|\,c)}{P_r(\mathrm{disease}\,|\,c)}, \tag{34}$$

where $P_r(\mathrm{disease}\,|\,c)$ is the 'baseline' risk, that is, the probability of disease of a random embryo ($r$ stands for *random*; this can also be seen as the risk in natural procreation). Note that we can similarly define the absolute risk reduction (ARR) as $P_r(\mathrm{disease}\,|\,c) - P_s(\mathrm{disease}\,|\,c)$.

We have already computed $P_s(\mathrm{disease}\,|\,c)$ for the two selection strategies (*Equations (23) and (29)*). To compute $P_r(\mathrm{disease}\,|\,c)$, we write the liability of a random embryo as

$$\begin{aligned} y\ &= s + e \\ &= x + c + e \\ &= \tilde{x} + c, \end{aligned} \tag{35}$$

where we defined $\tilde{x} = x + e$. $\mathrm{Var}(\tilde{x}) = \mathrm{Var}(x) + \mathrm{Var}(e) = r_{\mathrm{ps}}^2/2 + 1 - r_{\mathrm{ps}}^2 = 1 - r_{\mathrm{ps}}^2/2$, and thus, $\tilde{x} \sim N\left(0, 1 - r_{\mathrm{ps}}^2/2\right)$. The conditional probability of disease is

$$\begin{aligned} P_r(\mathrm{disease}\,|\,c)\ &= P(y > z_K\,|\,c) \\ &= P(\tilde{x} + c > z_K) \\ &= P(\tilde{x} > z_K - c) \\ &= 1 - \Phi\left(\frac{z_K - c}{\sqrt{1 - r_{\mathrm{ps}}^2/2}}\right). \end{aligned} \tag{36}$$

## 5.2 The distribution of the *per-couple* relative risk reduction

We can compute the probability density of $\mathrm{pcRRR}(c)$ across all couples in the population, $f_{pc}(x)$, as follows,

$$f_{pc}(x) = \int_{-\infty}^{\infty} \delta(x - \mathrm{pcRRR}(c)) f(c)\, dc, \tag{37}$$

where $\delta(x)$ is Dirac's delta function, $c$ is the parental average score, and $f(c) \sim N\left(0, r_{\mathrm{ps}}^2/2\right)$ is its

density. For computing $f_{pc}(x)$ numerically, we sum over $10^4$ quantiles of $c$ (which by definition have equal probability), and then compute the probability of the pcRRR to have value within each bin,

$$P(\text{pcRRR} \in [r_1, r_2]) = \frac{1}{10^4} \sum_{i=1}^{10^4} \boldsymbol{1}_{\text{pcRRR}(c_i) \in [r_1, r_2]}, \tag{38}$$

where $\boldsymbol{1}$ is the indicator variable, and $c_i$ is the $i/10^4$ quantile of $c$ (a value such that $c$ is less than $c_i$ with probability $(i - 0.5)/10^4$).

The average pcRRR across all couples is

$$\langle \text{pcRRR} \rangle = \int_{-\infty}^{\infty} \text{pcRRR}(c) f(c) dc. \tag{39}$$

Numerically,

$$\langle \text{pcRRR} \rangle = \frac{1}{10^4} \sum_{i=1}^{10^4} \text{pcRRR}(c_i). \tag{40}$$

Note that *Equation (39)* is an average of ratios. This is in contrast to *Equation (33)*, which a ratio of averages. As such, those average risk reductions are not expected to be identical. Empirically, given that $\text{pcRRR}(c)$ depends only weakly on $c$, we found that differences were small. For example, $\langle \text{pcRRR} \rangle$ was higher than the RRR from *Equation (33)* by $\approx 0.01$ for $r_{\text{ps}}^2 \leq 0.1$ (for $K = 0.01, 0.05, 0.2$); for example, when $K = 0.05$ and $r_{\text{ps}}^2 = 0.1$, $\langle \text{pcRRR} \rangle$ was 0.48, while the RRR was 0.47. Differences were larger for $r_{\text{ps}}^2 = 0.3$; for example, for $K = 0.05$, $\langle \text{pcRRR} \rangle$ was 0.77, while the RRR was 0.72.

## 5.3 The *per-batch* relative risk reduction

The pcRRR, that is, *Equation (34)*, can be interpreted as follows. A given couple can choose between two options: either generate embryos by IVF and select an embryo based on its PRS, or select an embryo at random (=conceive naturally). The pcRRR quantifies the risk reduction between the outcomes under these two choices. For each choice, the risk is computed by averaging over all possible embryos that might have been generated in an IVF cycle. However, one may also wish to quantify the variability of the outcome for a given couple. This could be accomplished as follows: for each couple *and* for each batch of $n$ embryos, compute the relative risk reduction when selecting an embryo based on PRS vs when selecting at random. We define this quantity as the *per-batch* relative risk reduction, or pbRRR.

Modeling the pbRRR is straightforward using our framework. Given the scores of the embryos, $s_1, \ldots, s_n$, the selected embryo is immediately determined for the *lowest-risk prioritization* strategy. For the *high-risk exclusion* strategy, the selected embryo can be, with equal probability, any of the embryos that are not high risk (or any embryo if all embryos are high risk). For random selection, the selected embryo can be any embryo with equal probability. Given the score of the selected embryo, $s_{i^*}$, and given the non-score component, $e \sim N(0, 1 - r_{\text{ps}}^2)$, the probability of disease of the selected embryo is

$$
\begin{aligned}
P_s(\text{disease}) &= P(y > z_K \,|\, s_{i^*}) \\
&= P(s_{i^*} + e > z_K) \\
&= P(e > z_K - s_{i^*}) \\
&= 1 - \Phi\left( \frac{z_K - s_{i^*}}{\sqrt{1 - r_{\text{ps}}^2}} \right).
\end{aligned}
\tag{41}
$$

The probability density of the scores is then given by *Equation (13)*. The distribution of the pbRRR across batches of embryos can then be computed by integrating over all possible sets of $n$ scores, similarly to *Equation (37)*. However, this would be tedious in practice, and we do not pursue this direction here.

## 6 The risk reduction conditional on family history

In the following, we compute the relative risk reduction when the disease status of the parents is given.

### 6.1 Model

Let us rewrite our model for the liability as

$$y = s + w + \epsilon. \tag{42}$$

Here, $w$ represents all genetic factors not included in the score. We keep track of both $s$ and $w$, because both are inherited, and hence, information on the disease status of the parents will be informative on their values in children (see below). However, we need to track each term separately because selection is only based on $s$. As in Section 1, we assume $s$, $w$, and $\epsilon$ are independent, $y \sim N(0,1)$, $s \sim N(0, r_{\mathrm{ps}}^2)$, and $\epsilon \sim N(0, 1-h^2)$, and thus $w \sim N(0, h^2 - r_{\mathrm{ps}}^2)$.

We derive the risk to the embryos in two main steps. First, we assume that the values of $s$ and $w$ are known for each parent, and compute the risk of the embryo under each selection strategy (*lowest-risk prioritization*, *high-risk exclusion*, or *random selection*). Then, we derive the posterior distribution of the parental genetic components given the parental disease status, and integrate over these components to obtain the final risk estimate.

### 6. 2 The risk of the selected embryo given its score

Denote the maternal score as $s_m$ and the paternal score as $s_f$, denote similarly $w_m$ and $w_f$, and assume that they are given. Also denote $g_m = s_m + w_m$ and $g_f = s_f + w_f$. As we explained in Section 2.1, for any child $i$, the distribution of the score $s_i$ is

$$s_i \sim N\left(\frac{s_m + s_f}{2}, \frac{r_{\mathrm{ps}}^2}{2}\right) \text{ or } s_i = c + x_i, \tag{43}$$

where $c = (s_m + s_f)/2$ and $x_i \sim N(0, r_{\mathrm{ps}}^2/2)$. Similarly, the distribution of the non-score genetic component is

$$w_i \sim N\left(\frac{w_m + w_f}{2}, \frac{h^2 - r_{\mathrm{ps}}^2}{2}\right) \text{ or } w_i = \frac{w_m + w_f}{2} + v_i, \tag{44}$$

where $v_i \sim N\left(0, (h^2 - r_{\mathrm{ps}}^2)/2\right)$.

Given the parental genetic components, we can write the liability of each embryo as, for $i = 1, \ldots, n$,

$$y_i = \frac{s_m + s_f}{2} + x_i + \frac{w_m + w_f}{2} + v_i + \epsilon_i, \tag{45}$$

where $\epsilon_i \sim N(0, 1-h^2)$. All the three random variables in the above equation ($x_i$, $v_i$, and $\epsilon_i$) are independent, and $x_i$ and $v_i$ are each independent across embryos. (It is not necessary to specify whether the $\epsilon_i$ are independent.) Denote the event that embryo $i$ is affected as $D_i$, and condition on the value of $x_i$ for that embryo. The probability of disease is

$$
\begin{aligned}
P\left(D_i \mid s_m, w_m, s_f, w_f, x_i\right) &= P(y_i > z_K \mid s_m, w_m, s_f, w_f, x_i) \\
&= P\left(\frac{s_m + s_f}{2} + x_i + \frac{w_m + w_f}{2} + v_i + \epsilon_i > z_K\right) \\
&= P\left(v_i + \epsilon_i > z_K - \frac{s_m + s_f}{2} - \frac{w_m + w_f}{2} - x_i\right) \\
&= 1 - \Phi\left(\frac{z_K - \frac{s_m + s_f}{2} - \frac{w_m + w_f}{2} - x_i}{\sqrt{1 - h^2/2 - r_{\mathrm{ps}}^2/2}}\right).
\end{aligned}
\tag{46}
$$

The last line holds because $\mathrm{Var}(v_i + \epsilon_i) = (h^2 - r_{\mathrm{ps}}^2)/2 + (1-h^2) = 1 - h^2/2 - r_{\mathrm{ps}}^2/2$.

We henceforth denote $D_s$ as the event that the selected embryo is affected. In the next three sub-sections, we integrate the probability of the disease over $x_i$, where the distribution of $x_i$ will vary depending on the selection strategy. This will give us the disease risk given the parental genetic components.

## 6.3 Selecting the lowest-risk embryo

Denote by $x_{i^*}$ the embryo-specific component of the embryo with the lowest such component. Recall that for each embryo, $x_i \sim N(0, r_{\text{ps}}^2/2)$. We can use the theory of order statistics, as in previous sections, to compute the density of $x_{i^*}$.

$$f(x_{i^*}) = \frac{n}{r_{\text{ps}}/\sqrt{2}} \phi\left(\frac{x_{i^*}}{r_{\text{ps}}/\sqrt{2}}\right)\left[1 - \Phi\left(\frac{x_{i^*}}{r_{\text{ps}}/\sqrt{2}}\right)\right]^{n-1}. \tag{47}$$

*Equation (46)* can now be integrated over all $x_{i*}$. After changing variables $t = x_{i^*}/(r_{\text{ps}}/\sqrt{2})$, we obtain

$$P(D_s \mid s_m, w_m, s_f, w_f) =$$
$$= \int_{-\infty}^{\infty} n\phi(t)[1 - \Phi(t)]^{n-1}\left[1 - \Phi\left(\frac{z_K - \frac{s_m + s_f}{2} - \frac{w_m + w_f}{2} - tr_{\text{ps}}/\sqrt{2}}{\sqrt{1 - h^2/2 - r_{\text{ps}}^2/2}}\right)\right]dt$$
$$= \int_{-\infty}^{\infty} n\phi(t)[1 - \Phi(t)]^{n-1}\left[1 - \Phi\left(\frac{z_K - \frac{g_m + g_f}{2} - tr_{\text{ps}}/\sqrt{2}}{\sqrt{1 - h^2/2 - r_{\text{ps}}^2/2}}\right)\right]dt. \tag{48}$$

Note that the final result depends only on $g_m$ and $g_f$. Thus, *Equation (48)* can be integrated over $g_m$ and $g_f$ (according to their posterior distribution given the family disease history; see Section 6.6) to provide the disease risk probability.

## 6.4 Excluding high-risk embryos

Here, the density of the score of the selected embryo is given by *Equation (28)*, which continues to hold, with $c = (s_m + s_f)/2$.

$$f(x_s) = \begin{cases} \frac{\frac{1}{r_{\text{ps}}/\sqrt{2}}\phi\left(\frac{x_s}{r_{\text{ps}}/\sqrt{2}}\right)}{1 - \Phi\left(\frac{z_q r_{\text{ps}} - c}{r_{\text{ps}}/\sqrt{2}}\right)}\left[1 - \Phi\left(\frac{z_q r_{\text{ps}} - c}{r_{\text{ps}}/\sqrt{2}}\right)\right]^n & \text{for } x_s > z_q r_{\text{ps}} - c \\[4mm] \frac{\frac{1}{r_{\text{ps}}/\sqrt{2}}\phi\left(\frac{x_s}{r_{\text{ps}}/\sqrt{2}}\right)}{\Phi\left(\frac{z_q r_{\text{ps}} - c}{r_{\text{ps}}/\sqrt{2}}\right)}\left\{1 - \left[1 - \Phi\left(\frac{z_q r_{\text{ps}} - c}{r_{\text{ps}}/\sqrt{2}}\right)\right]^n\right\} & \text{for } x_s < z_q r_{\text{ps}} - c \end{cases} \tag{49}$$

Integrating over all $x_s$, following similar steps as in Section 4, we obtain, denoting by $D_s$ the event that the selected embryo is affected,

$$P(D_s \mid s_m, w_m, s_f, w_f) = \int_{-\infty}^{\infty} \eta(t, \gamma)\xi(t)dt, \tag{50}$$

where we defined

$$\begin{aligned} \xi(t) &= \phi(t)\left[1 - \Phi\left(\frac{z_K - tr_{\text{ps}}/\sqrt{2} - \frac{s_m + s_f}{2} - \frac{w_m + w_f}{2}}{\sqrt{1 - h^2/2 - r_{\text{ps}}^2/2}}\right)\right] \\ &= \phi(t)\left[1 - \Phi\left(\frac{z_K - tr_{\text{ps}}/\sqrt{2} - \frac{g_m + g_f}{2}}{\sqrt{1 - h^2/2 - r_{\text{ps}}^2/2}}\right)\right], \\ \eta(t, \gamma) &= \begin{cases} \frac{1 - [1 - \Phi(\gamma)]^n}{\Phi(\gamma)} & \text{for } t < \gamma, \\ [1 - \Phi(\gamma)]^{n-1} & \text{for } t > \gamma \end{cases}, \text{ and} \\ \gamma &= \sqrt{2}z_q - \frac{c}{r_{\text{ps}}/\sqrt{2}}. \end{aligned} \tag{51}$$

Here, **Equation (50)** depends on $c, g_m, g_f$, and they must be integrated over to obtain the final disease probability.

## 6.5 The baseline risk

To compute the relative risk reduction, we need the baseline risk, that is, the risk when selecting a embryo at random given the parental genetic components. We have

$$
\begin{aligned}
P\left(D_s \mid s_m, w_m, s_f, w_f\right) &= P(y_i > z_K) \\
&= P\left(\frac{s_m + s_f}{2} + x_i + \frac{w_m + w_f}{2} + v_i + \epsilon_i > z_K\right) \\
&= P\left(x_i + v_i + \epsilon_i > z_K - \frac{g_m + g_f}{2}\right) \\
&= 1 - \Phi\left(\frac{z_K - \frac{g_m + g_f}{2}}{\sqrt{1 - h^2/2}}\right).
\end{aligned}
\tag{52}
$$

The last line holds because $\mathrm{Var}(x_i + v_i + \epsilon_i) = r_{\mathrm{ps}}^2/2 + (h^2 - r_{\mathrm{ps}}^2)/2 + (1 - h^2) = 1 - h^2/2$.

## 6.6 The disease risk conditional on the parental disease status

In subsections 6.3, 6.4, and 6.5, we computed the disease probability under the various strategies given the parental genetic components. For the baseline risk and for the *lowest-risk prioritization* strategy, the risk depended only on $g_m$ and $g_f$. For the *high-risk exclusion* strategy, the risk also depended on $c$. In this section, we compute the posterior probability of these genetic components conditional on the disease status of the parents.

Denote by $D_m$ the indicator variable that the mother is affected (i.e. $D_m = 1$ if the mother is affected and $D_m = 0$ otherwise), and define $D_f$ similarly. The risk of the selected embryo conditional on the parental disease status can be written as

$$
\begin{aligned}
P\left(D_s \mid D_m, D_f\right) &= \iiint dg_m dg_f dc\, P\left(D_s \mid g_m, g_f, c, D_m, D_f\right) f\left(g_m, g_f, c \mid D_m, D_f\right) \\
&= \iiint dg_m dg_f dc\, P\left(D_s \mid g_m, g_f, c\right) f\left(c \mid g_m, g_f\right) f\left(g_m, g_f \mid D_m, D_f\right).
\end{aligned}
\tag{53}
$$

The second line of **Equation (53)** consists of three terms. The first is $P\left(D_s \mid g_m, g_f, c\right)$, which was computed in the previous subsections for the various selection strategies. Note that we assumed $P\left(D_s \mid g_m, g_f, c, D_m, D_f\right) = P\left(D_s \mid g_m, g_f, c\right)$. This holds because given the genetic components of the parents, their disease status does not provide additional information on the disease status of the children, at least under a model where the environment is not shared (see Section 10). The second term is the density of $c$, which can be similarly written as $f\left(c \mid g_m, g_f, D_m, D_f\right) = f\left(c \mid g_m, g_f\right)$. The third term is the posterior distribution of $g_m$ and $g_f$ given the parental disease status, $f\left(g_m, g_f \mid D_m, D_f\right)$. In the following, we derive the third term and then the second term.

Note that if $P\left(D_s \mid g_m, g_f, c\right) = P\left(D_s \mid g_m, g_f\right)$, as in the case of the baseline risk (**Equation (52)**) and the *lowest-risk prioritization* (**Equation (48)**), the risk of the selected embryo can be simplified by integrating over $c$,

$$
P\left(D_s \mid D_m, D_f\right) = \iint dg_m dg_f\, P\left(D_s \mid g_m, g_f\right) f\left(g_m, g_f \mid D_m, D_f\right).
\tag{54}
$$

## 6.7 The distribution of the parental genetic components given the parental disease status

First, we assume (given that we did not model assortative mating) that given one parent's disease status, his/her genetic component is independent of the spouse's disease status or genetic factors. Thus, the posterior distribution can be factored into

$$f(g_m, g_f \mid D_m, D_f) = f(g_m \mid D_m) f(g_f \mid D_f). \tag{55}$$

Next, without loss of generality, we focus on just the mother. To derive the posterior distribution $f(g_m \mid D_m)$, we first need the prior, $g_m \sim N(0, h^2)$.

$$f_{pr}(g_m) = \frac{1}{h} \phi\left(\frac{g_m}{h}\right). \tag{56}$$

Next, the likelihood that the mother is affected is

$$P(D_m = 1 \mid g_m) = P(y > z_K) = P(g_m + \epsilon > z_K) = P(\epsilon > z_K - g_m) = 1 - \Phi\left(\frac{z_K - g_m}{\sqrt{1 - h^2}}\right). \tag{57}$$

Similarly,

$$P(D_m = 0 \mid g_m) = \Phi\left(\frac{z_K - g_m}{\sqrt{1 - h^2}}\right). \tag{58}$$

Using Bayes' theorem,

$$f(g_m \mid D_m = 1) = \frac{P(D_m = 1 \mid g_m) f_{pr}(g_m)}{P(D_m = 1)} = \frac{\left[1 - \Phi\left(\frac{z_K - g_m}{\sqrt{1 - h^2}}\right)\right] \frac{1}{h} \phi\left(\frac{g_m}{h}\right)}{K}. \tag{59}$$

Similarly,

$$f(g_m \mid D_m = 0) = \frac{P(D_m = 0 \mid g_m) f_{pr}(g_m)}{P(D_m = 0)} = \frac{\Phi\left(\frac{z_K - g_m}{\sqrt{1 - h^2}}\right) \frac{1}{h} \phi\left(\frac{g_m}{h}\right)}{1 - K}. \tag{60}$$

The same results hold for $f(g_f \mid D_f = 1)$ and $f(g_f \mid D_f = 1)$. We have thus specified the posterior distribution $f(g_m, g_f \mid D_m, D_f)$.

## 6.8 The distribution of the parental mean score given the parental genetic components

The final missing term is $f(c \mid g_m, g_f)$. To compute this distribution, we note that $c$, $g_m$, and $g_f$ have a multivariate normal distribution,

$$(c, g_m, g_f) \sim \mathrm{MVN}\left(\begin{pmatrix} 0 \\ 0 \\ 0 \end{pmatrix}, \begin{pmatrix} \frac{r_{ps}^2}{2} & \frac{r_{ps}^2}{2} & \frac{r_{ps}^2}{2} \\ \frac{r_{ps}^2}{2} & h^2 & 0 \\ \frac{r_{ps}^2}{2} & 0 & h^2 \end{pmatrix}\right). \tag{61}$$

To explain the above equation, recall that $\mathrm{Var}(c) = r_{ps}^2 / 2$ and $\mathrm{Var}(g_m) = \mathrm{Var}(g_f) = h^2$. Then,

$$\mathrm{Cov}(c, g_m) = \mathrm{Cov}\left(\frac{s_m + s_f}{2}, g_m\right) = \frac{1}{2} \mathrm{Cov}(s_m, g_m) = \frac{1}{2} \mathrm{Cov}(s_m, s_m + w_m) = \frac{1}{2} \mathrm{Var}(s_m) = \frac{r_{ps}^2}{2}. \tag{62}$$

A similar result holds for the paternal genetic component. To compute the density of $c$ given $g_m$ and $g_f$, we use standard theory for multivariate normal variables (as in Section 2.1). We have

$$c \mid g_m, g_f \sim N(\mu, \sigma^2), \tag{63}$$

with

$$\mu = \frac{r_{ps}^2}{h^2}\left(\frac{g_m + g_f}{2}\right), \sigma^2 = \frac{r_{ps}^2}{2h^2}(h^2 - r_{ps}^2). \tag{64}$$

We have thus specified $f(c \mid g_m, g_f)$.

## 6.9 Summary of the computation

In summary, for the *high-risk exclusion* strategy, the probability of disease of the selected embyro given the parental disease status is given by *Equation (53)*, with $P(D_s \mid g_m, g_f, c)$ given in *Equation (50)* and $f(c \mid g_m, g_f)$ in *Equation (63)*. The conditional probability of disease for the *lowest-risk prioritization* strategy and for random selection (the baseline risk) is given by *Equation (54)*, with $P(D_s \mid g_m, g_f)$ given in *Equations (48)* and *Equation (52)*, respectively. For all selection strategies, $f(g_m, g_f \mid D_m, D_f)$ is given by *Equations (55), (59), and (60)*, depending on the particular family history.

Numerically, computing the baseline disease risk requires two integrals (over $g_m$ and $g_f$). Computing the risk for the *lowest-risk prioritization* strategy requires three integrals (over $g_m$, $g_f$, and $t$). Computing the risk for the *high-risk exclusion* strategy requires four integrals (over $g_m$, $g_f$, $c$, and $t$).

## 7 Two diseases

Prioritizing embryos based on low risk for a target disease may increase risk for a second disease, if that disease is genetically anti-correlated with the target disease. In this section, we develop a model for the PRSs of two diseases in order to investigate this risk.

We denote the variance explained by the scores of the two diseases as $r_1^2$ and $r_2^2$, where disease 1 is the target disease (i.e. embryos are prioritized based on their risk for that disease), and disease 2 is the correlated disease. Denote the genetic correlation between the diseases as $\rho$ (where $\rho<0$ is the case raising the concern about increasing the risk of the correlated disease), the scores of a child as $s^{(1)}$ and $s^{(2)}$, the scores of the mother as $s_m^{(1)}$ and $s_m^{(2)}$, and the scores of the father as $s_f^{(1)}$ and $s_f^{(2)}$. The vector $(s^{(1)}, s^{(2)}, s_m^{(1)}, s_m^{(2)}, s_f^{(1)}, s_f^{(2)})$ has a multivariate normal distribution, with zero means, and with the following covariance matrix (extending *Equation (9)*).

$$
\Sigma = \begin{array}{c} \\ s^{(1)} \\ s^{(2)} \\ s_m^{(1)} \\ s_m^{(2)} \\ s_f^{(1)} \\ s_f^{(2)} \end{array}
\begin{pmatrix}
r_1^2 & \rho r_1 r_2 & \frac{r_1^2}{2} & \frac{\rho r_1 r_2}{2} & \frac{r_1^2}{2} & \frac{\rho r_1 r_2}{2} \\
\rho r_1 r_2 & r_2^2 & \frac{\rho r_1 r_2}{2} & \frac{r_2^2}{2} & \frac{\rho r_1 r_2}{2} & \frac{r_2^2}{2} \\
\frac{r_1^2}{2} & \frac{\rho r_1 r_2}{2} & r_1^2 & \rho r_1 r_2 & 0 & 0 \\
\frac{\rho r_1 r_2}{2} & \frac{r_2^2}{2} & \rho r_1 r_2 & r_2^2 & 0 & 0 \\
\frac{r_1^2}{2} & \frac{\rho r_1 r_2}{2} & 0 & 0 & r_1^2 & \rho r_1 r_2 \\
\frac{\rho r_1 r_2}{2} & \frac{r_2^2}{2} & 0 & 0 & \rho r_1 r_2 & r_2^2
\end{pmatrix}
\tag{65}
$$

In the above covariance matrix, we assumed that the correlation between the scores of the two diseases is also $\rho$. The covariance between parent-child scores for different diseases is half the covariance of the scores within an individual (e.g. see *Karavani et al., 2019*).

Next, we need the density of $(s^{(1)}, s^{(2)})$, conditional on $(s_m^{(1)}, s_m^{(2)}, s_f^{(1)}, s_f^{(2)})$. We follow a similar procedure as in Section 2.1, and obtain the conditional density of $(s^{(1)}, s^{(2)})$ as $\mathrm{MVN}(\mu_c, \Sigma_c)$ with

$$
\mu_c = \begin{pmatrix} \frac{s_m^{(1)} + s_f^{(1)}}{2} \\ \frac{s_m^{(2)} + s_f^{(2)}}{2} \end{pmatrix}, \Sigma_c = \begin{pmatrix} \frac{r_1^2}{2} & \frac{\rho r_1 r_2}{2} \\ \frac{\rho r_1 r_2}{2} & \frac{r_2^2}{2} \end{pmatrix}.
\tag{66}
$$

We would like to compute the expected increase in risk to become affected by the second disease, given any selection strategy of embryos based on a PRS for the first disease. Solving this problem analytically is beyond the scope of this work. However, the above results imply a method we could use for simulations.

Let us first consider how to draw the average parental scores, which we denote $c^{(1)} = \left(s_m^{(1)} + s_f^{(1)}\right)/2$ and $c^{(2)} = \left(s_m^{(2)} + s_f^{(2)}\right)/2$. The vector $(c^{(1)}, c^{(2)})$ has a multivariate normal distribution with zero means (as each parental score has zero mean in the population), and the following covariance matrix. The variances are $\mathrm{Var}(c^{(1)}) = r_1^2/2$ and $\mathrm{Var}(c^{(2)}) = r_2^2/2$. The covariance is

$$\mathrm{Cov}\left(c^{(1)}, c^{(2)}\right) = \mathrm{Cov}\left(\frac{s_m^{(1)} + s_f^{(1)}}{2}, \frac{s_m^{(2)} + s_f^{(2)}}{2}\right) = \frac{1}{2}\mathrm{Cov}\left(s_m^{(1)}, s_m^{(2)}\right) = \frac{\rho r_1 r_2}{2}. \tag{67}$$

Thus, the covariance matrix is equal to $\Sigma_c$ from *Equation (66)* above. This suggests the following simple algorithm for generating the risk scores of the embryos.

$$\begin{pmatrix} c^{(1)} \\ c^{(2)} \end{pmatrix} \sim \mathrm{MVN}\left(\begin{pmatrix} 0 \\ 0 \end{pmatrix}, \begin{pmatrix} \frac{r_1^2}{2} & \frac{\rho r_1 r_2}{2} \\ \frac{\rho r_1 r_2}{2} & \frac{r_2^2}{2} \end{pmatrix}\right), \begin{pmatrix} s_i^{(1)} \\ s_i^{(2)} \end{pmatrix} \sim \mathrm{MVN}\left(\begin{pmatrix} c^{(1)} \\ c^{(2)} \end{pmatrix}, \begin{pmatrix} \frac{r_1^2}{2} & \frac{\rho r_1 r_2}{2} \\ \frac{\rho r_1 r_2}{2} & \frac{r_2^2}{2} \end{pmatrix}\right), \ i = 1, \dots, n. \tag{68}$$

In our simulations, we select an embryo based on its score for disease 1, according to a selection strategy. We draw the non-score component for disease 1 of the selected embryo as $e^{(1)} \sim N(0, 1 - r_1^2)$, and the liability of the embryo for that disease is then $y^{(1)} = s^{(1)} + e^{(1)}$. We draw the liability for disease 2 of the selected embryo similarly. In our simulations, we draw $e^{(1)}$ and $e^{(2)}$ independently, even though they are correlated. [Any genetic component not included in the score should be correlated between the diseases, and the environments affecting the two diseases may be correlated as well.] We do this because we are only interested in the marginal outcome for each disease separately (i.e. not in the joint outcome of the two diseases). The selected embryo is designated as affected by each disease if the liability of that disease exceeds its respective threshold.

We note that the above model represents the following approximation. As the scores are (noisily) estimating the total genetic effects, the score of one disease is correlated with the non-score genetic component of the other disease. Thus, a more accurate expression for the liability to disease 2 would take into account not only $s_i^{(2)}$ but also $s_i^{(1)}$. However, the dependence is expected to be weak.

## 8 Comparison to previous work

In the 'gwern' blog (*Gwern, 2018*), the utility of embryo selection for traits and/or diseases was investigated. For disease risk, a model similar to ours was studied, based on the liability threshold model. However, the model assumed that given the polygenic score, the distribution of the remaining contribution to the liability has unit variance, instead of $1 - r_{\mathrm{ps}}^2$ (the function liabilityThresholdValue therein). Further, the blog provided only numerical results, and it did not consider the variability in the gain, the *high-risk exclusion* strategy, the risk reduction conditional on the parental scores or disease status, and the *per-couple* relative risk reduction. Treff et al. (*Treff et al., 2019a*) also employed the liability threshold model to evaluate embryo selection for disease risk. However, they did not consider the *high-risk exclusion* strategy, and did not compute analytically the risk reduction. They only provided simulation results for the case when a parent is affected based on an approximate model.

## 9 Simulations

Our analytical results in the above sections provide exact expressions for the relative risk reduction under various settings in the form of integrals, which we then solve numerically. To validate our analytical derivations and the numerical solutions, we also simulated the scores of embryos under each setting, and verified that the empirical risk reductions agree with the analytical predictions.

To simulate the scores of embryos, we used the representation $s_i = x_i + c$, where $(x_1, \dots, x_n)$ are independent normals with zero means and variance $r_{\mathrm{ps}}^2/2$, and $c \sim N(0, r_{\mathrm{ps}}^2/2)$ is shared across all embryos. Thus, for each 'couple', we first draw $c$, then draw $n$ independent normals $(x_1, \dots, x_n)$, and then compute the score of embryo $i$ as $s_i = x_i + c$, for $i = 1, \dots, n$. The score of the selected embryo was the lowest among the $n$ embryos in the *lowest-risk prioritization* strategy. For the *high-risk exclusion* strategy, we selected the first embryo with score $s < z_q r_{\mathrm{ps}}$. If no such embryo existed, we selected the first embryo (except for one analysis, in which, if all embryos were high-risk, we selected the embryo with the lowest score). We then drew the residual of the liability as $e \sim N(0, 1 - r_{\mathrm{ps}}^2)$, and computed the liability as $s^* + e$, where $s^*$ is the score of the selected embryo. If the liability exceeded the threshold $z_K$, we designated the embryo as affected. We repeated over $10^6$ couples, and computed

the probability of disease as the fraction of couples in which the selected embryo was affected. We computed the relative risk reduction using *Equation (33)*.

For the setting when the parental risk scores are given, we computed $c$ as $c = (s_m + s_f)/2$. We specified the maternal score as a percentile $p_m$, such that the score itself was $s_m = z_{p_m} r_{ps}$, where $z_{p_m}$ is the $p_m$ percentile of the standard normal distribution. We similarly specified the paternal score. The remaining calculations were as above. For the baseline risk, we used the same data, assuming that the first embryo in each family was selected.

When conditioning on the parental disease status, we first drew the three independent parental components, all as normal variables with zero mean. We drew $s_m$ and $s_f$ with variance $r_{ps}^2$; $w_m$ and $w_f$ with variance $h^2 - r_{ps}^2$; and $\epsilon_m$ and $\epsilon_f$ with variance $1 - h^2$. We computed the maternal liability as $y_m = s_m + w_m + \epsilon_m$, and designated the mother as affected if $y_m > z_K$. We similarly assigned the paternal disease status. We then drew the score of each embryo as $s_i = c + x_i$, where $c = (s_m + s_f)/2$ (using the already drawn parental scores) and $x_i \sim N(0, r_{ps}^2/2)$, for $i = 1, \ldots, n$, are independent across embryos. We selected one embryo based on the selection strategy, as described above. If $s^*$ is the score of the selected embryo, we computed the liability of the selected embyro as $s^* + (w_m + w_f)/2 + v + \epsilon$, where $v \sim N(0, (h^2 - r_{ps}^2)/2)$ and $\epsilon \sim N(0, 1 - h^2)$. We designated the embryo as affected if its liability exceeded $z_K$. We tallied the proportion of affected embryos separately for each number of affected parents (0,1, or 2). To compute the baseline risk, we again used the first embryo in each family.

For two diseases, we do not have an analytical solution for the change in risk of the second disease. We thus evaluated the risk using simulations only. We considered the *lowest-risk prioritization* strategy and the case of random parents. For each couple and for each embryo, we generated polygenic scores for the two diseases as outlined in Section 7. We selected the embryo with the lowest score for the target disease, but then considered the score of that embryo for the second, correlated disease. Denote by $s^{*(2)}$ the score of the selected embryo for the second disease. We drew the residual of the liability for the second disease as $e^{(2)} \sim N(0, 1 - r_2^2)$, and the liability of the embryo for that disease was then $s^{*(2)} + e^{(2)}$. If the liability exceeded the threshold of that disease, we designated the embryo as affected. We also repeated for a random selection of an embryo for each couple. We computed the relative risk increase based on the ratio between the risks with or without PRS-based selection.

## 10 Limitations of the model

Our model has a number of limitations. First, our results rely on several modeling assumptions. (1) We assumed an infinitesimal genetic architecture for the disease, which will not be appropriate for oligogenic diseases or when screening the embryos for variants of very large effect. We did not assess the robustness of our theoretical results to deviations from normality in the tails of the distributions of the genetic and non-genetic components (although the good agreement with the simulations based on the real genomic data provide some support that the model is reasonable). (2) Assumption (1) implies that the variance of the scores of children is always half the population variance, regardless of the value of the parental PRSs or the disease considered (*Equation 13*). However, as shown in *Chen et al., 2020*, the variance of the scores in children can vary across families. On the other hand, Chen et al. also showed (*Figure 3C* therein) that between-family differences decrease when increasing the number of variants included in the PRS; and, as we showed here, the differences seem to be explained mostly by sampling variance. (3) Our model also assumes no assortative mating, which seems reasonable given that for genetic disease risk, correlation between parents is weak (*Rawlik et al., 2019*), and given that our previous study of traits showed no difference in the results between real and random couples (*Karavani et al., 2019*). (4) When conditioning on the parental disease status, we assumed independence between the environmental component of the child and either genetic or environmental factors influencing the disease status of the parents. Family-specific environmental factors were shown to be small for complex diseases (*Wang et al., 2017*). The influence of parental genetic factors on the child's environment is discussed in the next paragraph. Both of these influences, to the extent that they are significant, are expected to reduce the degree of risk reduction.

Second, we assumed that the proportion of variance (on the liability scale) explained by the score is $r_{\mathrm{ps}}^2$, but we did not specify how to estimate it. Typically, $r_{\mathrm{ps}}^2$ is computed and reported by large GWASs based on an evaluation of the score in a test set. However, the variance that will be explained by the score in other cohorts, using other chips, and particularly, in other populations, can be substantially lower (*Martin et al., 2019*). Relatedly, the variance explained by the score, as estimated in samples of unrelated individuals, is inflated due to population stratification, assortative mating, and indirect parental effects ('genetic nurture') (*Kong et al., 2018*; *Young et al., 2019*; *Morris et al., 2020*; *Mostafavi et al., 2020*), where the latter refers to trait-modifying environmental effects induced by the parents based on their genotypes. These effects do not contribute to prediction accuracy when comparing polygenic scores between siblings (as when screening IVF embryos), and thus, the variance explained by polygenic scores in this setting can be substantially reduced, in particular for cognitive traits. However, recent empirical work on within-family disease risk prediction showed that the reduction in accuracy is at most modest (*Lello et al., 2020*), and within-siblings-GWAS yielded similar results to unrelated-GWAS for most physiological traits (*Howe et al., 2021*).

Third, we implicitly assumed that polygenic scores could be computed with perfect accuracy based on the genotypes of IVF embryos. However, embryos are genotyped based on DNA from a single or very few cells, and whole-genome amplification results in high rates of allele dropout. Further, embryos are often sequenced to low depth. However, we and others have shown that very accurate genotyping of IVF embryos is feasible (*Backenroth et al., 2019*; *Kumar et al., 2015*; *Natesan et al., 2014*; *Treff et al., 2019b*; *Xiong et al., 2019*; *Yan et al., 2015*; *Zamani Esteki et al., 2015*). Either way, even if sequencing errors do occur, their effect can be readily taken into account. Suppose that $r_0^2$ is the proportion of variance in liability explained by a perfectly genotyped PRS, and that $r_{impute}^2$ is the squared correlation between the true score and the imputed score of an embryo (which can be estimated experimentally). Then $r_{\mathrm{ps}}^2 = r_0^2 \cdot r_{impute}^2$, where $r_{\mathrm{ps}}^2$ is the variance explained by the observed score, that is, the index used in our models.

Fourth, we did not model the process of IVF and the possible reasons for loss of embryos. Rather, we assumed that $n$ viable embryos are available that would have led to live birth if implanted. The original number of fertilized oocytes would typically be greater than $n$ (see, e.g. the 'gwern' blog for more detailed modeling). Similarly, we did not model the age-dependence of the number of embryos; again, we rather assume $n$ viable embryos are available. Finally, we assumed a single embryo transfer. In principle, transfer of, for example, two embryos is straightforward to simulate: we can select two embryos based on the selection strategy (e.g., under *lowest-risk prioritization*, select the two embryos with the lowest PRSs). Then, if only one of them is born, we can assume that the child is each of the embryos with probability 0.5. We expect the RRR to somewhat decrease under multiple embryo transfer, for both the *lowest-risk prioritization* and *high-risk exclusion* selection strategies. However, an analytical derivation seems difficult.

Fifth, the residual $e$ in *Equation (2)* ($y = s + e$) has a complex pattern of correlation between siblings. As noted in Section 1, $e$ has contributions from both genetic and environmental factors. The genetic covariance between siblings is straightforward to model (as in Section 2). However, the proportion of variance in liability explained by shared environment needs to be estimated and can be large (*Lakhani et al., 2019*). Further, embryos from the same IVF cycle (when only one is actually implanted) would have experienced the same early developmental environment and are thus expected to share even more environmental factors, similarly to twins. In the current work, the correlation between non-genetic factors across embryos does not enter our derivations. However, care must be taken in any attempt to model the joint phenotypic outcomes of multiple embryos.

Finally, in this work, we modeled various scenarios for the ascertainment of the parents: either randomly, or based on their scores, or based on their disease status. In future work, it will be interesting to model other settings of family history, such as the presence of an affected child. Further, it is likely that parents will attempt to screen the embryos for more than one disease (*Treff et al., 2020*). In future work, it will be important to model screening for multiple diseases and compute the expected outcomes.

## 11 Code availability

The R code we used to implement all modeling in this paper and generate the corresponding figures can be found at https://github.com/scarmi/embryo_selection.

To give two examples, below is an R function that computes the relative risk reduction under the *lowest-risk prioritization* strategy for randomly ascertained parents.

```
library(MASS)
risk_reduction_lowest = function(r,K,n)
{
  zk = qnorm(K, lower.tail=F)
  integrand_lowest = function(t)
   return(dnorm(t)*pnorm((zk-t*sqrt(1-r^2/2)) / (r/sqrt(2)), lower.tail=F)^n)
  risk = integrate(integrand_lowest,-Inf,Inf)$value
  return((K-risk)/K)
}
```

The R function below computes the relative risk reduction under the high-risk exclusion strategy (for randomly ascertained parents).

```
risk_reduction_exclude = function(r,K,q,n)
{
  zk = qnorm(K, lower.tail=F)
  zq = qnorm(q, lower.tail=F)
  integrand_t = function(t,u)
   return(dnorm(t)*pnorm((zk-r/sqrt(2)*(u+t))/sqrt(1-r^2),lower.tail=F))
  integrand_u = function(us)
  {
   y = numeric(length(us))
   for (i in seq_along(us))
   {
     u = us[i]
     beta = zq*sqrt(2)-u
     internal_int1 = integrate(integrand_t,-Inf,beta,u)$value
     denom1 = pnorm(beta)
     if (denom1==0) {denom1=1e-300} # Avoid dividing by zero
     numer1 = 1-pnorm(beta,lower.tail=F)^n
     internal_int2 = integrate(integrand_t,beta,Inf,u)$value
     prefactor2 = pnorm(beta,lower.tail=F)^(n-1)
     y[i] = dnorm(u) * (numer1/denom1*internal_int1 + prefactor2*internal_int2)
   }
   return(y)
  }
  risk = integrate(integrand_u,-Inf,Inf)$value
  return((K-risk)/K)
}
```

