## [Decision Letter]

**Acceptance summary:**

This work represents a thought-provoking and timely exploration of the impact of polygenic embryo screening using a liability threshold model. This work is expected to catalyse further debates in this emerging field.

**Decision letter after peer review:**

Thank you for submitting your article "Utility of polygenic embryo screening for disease depends on the selection strategy" for consideration by *eLife*. Your article has been reviewed by 3 peer reviewers, one of whom is a member of our Board of Reviewing Editors, and the evaluation has been overseen by Mone Zaidi as the Senior Editor. The following individual involved in review of your submission has agreed to reveal their identity: Qiongshi Lu (Reviewer #3).

Summary:

Polygenic embryo screening is a controversial topic. Due to generally low predictive accuracy of current polygenic risk scores (PRS), the entanglement of genetics's and environment's roles in disease etiology, and a lack of empirical studies, our understanding of potential benefits, and importantly, risks of PRS-based human embryo screening is qualitative at best and far from complete. In this study, Lencz et al. provide a statistical framework to assess the effectiveness of various screening strategies to reduce disease risk. Overall, due to some limitations in the study, the current results do not necessarily endorse practicing human embryo screening using PRS. Instead, the major contribution of this study is to layout the quantitative arguments and metrics which future studies may continue to use. These tools are crucial for researchers to understand the risk/benefit trade-off and to design better strategies. This work is thought-provoking and should open up useful debates in the field.

Major comments:

1. The main conclusions seem to contradict the same group's previous work (Karavani et al., 2019) which argued that screening embryos with PRS has limited utility. The key difference between this paper and the previous study is that here the focus is binary disease outcomes while the previous study focused on quantitative traits. However, the distinction between quantitative traits and their dichotomized counterparts is not always clear (e.g., cholesterol levels and hypercholesterolemia, education years and college attainment), which seems to suggest that the assessment of a screening strategy strongly depends on how the phenotype of interest is defined. Do the conclusions in (Karavani et al., 2019) still hold given the findings of this paper? Would it be fair to say that (i) given all assumptions made in this paper such as a lack of genetic nurture effects and rare variant effects, and (ii) if the interest is to prevent intellectual disability which is a dichotomized version of IQ, a trait extensively studied in Karavani et al. 2019, then polygenic embryo screening may NOT have limited utility as long as the low-risk prioritization approach is used? More extensive discussions are needed to compare the approaches and conclusions between the two closely related papers.

2. It appears to be counter-intuitive that the curves of the "high-risk exclusion" strategy would peak instead of being monotone (e.g. the upper-left panel in figure 2). If the exclusion percentile (i.e., x-axis) keeps increasing, shouldn't this approach behave similarly compared to the "low-risk prioritization" strategy? One would think that a stringent percentile threshold in "high-risk exclusion" (excluding most embryos) would ensure that only embryos with very low risks are planted, which is what the "low-risk prioritization" strategy tries to achieve. So why doesn't the purple curve in that upper-left panel keep going up until reaching ~80% which is suggested by the lower-left panel in Figure 2?

3. The authors should comment on the following points. First, partitioning the PRS "s_i_" into two orthogonal components: polygenic transmission disequilibrium "x_i_" and parental average score "c" seems intuitive. However, shouldn't parental genotypes predict the distribution of x_i_? Suppose both parents are homozygous at a SNP and c = (2+2)/2 = 2. Then, s_i_ would be 2 without any variability, which means x_i_ is always 0. Even in a polygenic setting, a very high value of c would suggest higher homozygosity which should lead to lower variance of x_i_. However, the setting in this paper leads to a constant variance of x_i_ which isn't affected by c. Is something missing here?

4. The authors state that they have not explored the age dependence on the number of embryos? How significant would this omission be? What is the age distribution of parents who would typically undertake such polygenic embryo selection?

5. For the high risk exclusion strategy, the authors state that if all embryos are of high risk, then a random embryo would be implanted. Would a decision of not implanting any embryo be a reasonable alternative? If this is indeed a reasonable alternative, would this change the authors' conclusion in any way?

6. The authors should clarify the impact of error in the per-embryo estimation of x_s_ (e.g. from errors in imputation of genotypes if that's used or errors in the β_hat estimates that form the PRS)? I assume in these cases it results in an effective lowering of rps2?7. While the statistical framework in this paper can be used to compare the utility of different strategies, the same analyses can also quantify the potential risks of embryo screening. Although the authors were open about the limitations of the study, no empirical analyses were performed to demonstrate potential risks. This is a missed opportunity. Can the authors expand this somewhat?

8. While it is understandable to make some simplifying assumptions, it is critical to demonstrate the risk which is a main concern of human embryo screening in the field. While it does not appear that the authors have over-stated anything in the original submission, given the sensitivity of this topic, avoiding quantitative discussions on risks may lead to over-optimistic news headlines that only focus on the potential utility and even reckless and potentially harmful practices. With that said, it seems straightforward to generalize the simulation framework in this paper to explore a two-trait setting. Suppose two diseases are genetically correlated (with correlation = -0.1, -0.3, and -0.5, for example). Then, would the "low-risk prioritization approach" on one disease lead to drastically increased risks for the second disease? It wouldn't be fair to ask the authors to find a solution to this problem given the scope of the study. But a quantitative demonstration of risks will improve this paper.

9. What is the current success rate if only one embryo is planted? If multiple embryos (out of 5) are planted to ensure a reasonable success rate in current practices, would it be fair to say that only a "high-risk exclusion"-type strategy that only excludes embryos with higher risk is realistic? Here, the exclusion criterion isn't necessarily based on a population quantile but could be based on PRS ranking among embryos. Some discussions will be helpful.

10. The authors state that one concern about the generalisability of polygenic embryo selection is the extension to non-European populations. Outside of the European populations, is it true that none of the other ethnic groups have high quality polygenic risk score data?

11. The usage of an extended listing of assumptions deep in the Material and Methods was slightly unorthodox to me and I wonder if it would be more transparent and useful for the average reader's consumptions to include these in the discussion itself. While the presented results are helpful, it's important that stakeholders digesting these numbers understand how many unknowns there are to still understand in this area.

12. Beyond the average risk reduction, would it be helpful to understand the variance in outcomes, e.g., which strategies have high variance in outcome? Can the authors speak to that dimension even qualitatively?

13. Perhaps one set of simulations with a finite and varying number of loci would be useful to guide intuition on when the infinitesimal model is appropriate. From experiences in other examples, the infinitesimal can already provide useful predictions for what might at first seem like small numbers of loci and if that holds in this case it might be useful for readers to understand who might otherwise be too quickly dismissive.

14. For clarity, please use a different variable besides in 'e' in sections where 'e' includes environment plus unmeasured genetic effects. Perhaps 'u' for "un-measured" effects?

15. Equation 16, e_i_* was not explicitly defined, even if it's inferable from context it would be nice to define.

---

## [Author Response]

Major comments:1. The main conclusions seem to contradict the same group's previous work (Karavani et al., 2019) which argued that screening embryos with PRS has limited utility. The key difference between this paper and the previous study is that here the focus is binary disease outcomes while the previous study focused on quantitative traits. However, the distinction between quantitative traits and their dichotomized counterparts is not always clear (e.g., cholesterol levels and hypercholesterolemia, education years and college attainment), which seems to suggest that the assessment of a screening strategy strongly depends on how the phenotype of interest is defined. Do the conclusions in (Karavani et al., 2019) still hold given the findings of this paper? Would it be fair to say that (i) given all assumptions made in this paper such as a lack of genetic nurture effects and rare variant effects, and (ii) if the interest is to prevent intellectual disability which is a dichotomized version of IQ, a trait extensively studied in Karavani et al. 2019, then polygenic embryo screening may NOT have limited utility as long as the low-risk prioritization approach is used? More extensive discussions are needed to compare the approaches and conclusions between the two closely related papers.

We thank the reviewers for raising this important point. The assessment of the reviewers is correct. Indeed, under the assumptions of the model (i.e., a polygenic architecture and the liability threshold model), it is possible to achieve large relative risk reductions for dichotomized traits. Before we discuss the rationale of this statement, we present numerical results (please see new Figure 2 —figure supplement 3).

Panel (A) of the figure considers IQ. We defined IQ<70 as “affected” (e.g., having an intellectual disability) and IQ>70 as “unaffected”. We varied the number of embryos n, and plotted the (theoretical) relative risk reduction when selecting the embryo with the highest score for IQ. We used rps2=0.052 (Savage et al., 2018). Panel (B) shows results for LDL cholesterol. Here, we fixed n=5 and varied the definition of “high LDL”. Specifically, we plotted the relative risk reduction when “high LDL” is defined as LDL-C value in the top 1%, 2%, 3%, ….,10% of the population. We assumed selection of the embryo with the lowest score, and rps2=0.12 (Weissbrod et al., 2021).

The results show that substantial relative risk reductions can be achieved, up to ≈40-50%, with just a small number of embryos, similarly to what we have described generally for diseases in our manuscript. But how do these results reconcile with our previous estimates of limited utility for continuous traits (Karavani et al., 2019)?

To understand the relation between continuous and binary traits, consider IQ. Our estimate for the mean gain in IQ that could be achieved by embryo screening has been ≈2.5 IQ points. Now assume, as above, that individuals with IQ<70 are considered “affected”. This implies that any hypothetical child that would have been born with IQ between 67.5 and 70 points will now have IQ above 70, and thus will be considered “unaffected”. Among individuals with IQ<70 (2 SDs below the mean), the proportion of individuals with IQ in the range [67.5,70] is 33.5% (assuming normal distribution). In other words, increasing IQ by merely 2.5 points is sufficient to drive a reduction of about a third in the probability to be “affected”. While this is an approximate calculation (as it does not take into account the variability in the gain), it demonstrates why even a small reduction in an underlying quantitative trait can substantially reduce the probability of extreme events.

Polygenic diseases behave similarly, when the underlying continuous trait is the liability. In other words, a small reduction in the liability will lead to a large reduction in the proportion of affected individuals. This is fundamentally a property of a threshold phenotype with an underlying normally distributed continuous liability. Most of the individuals in the extreme of the liability distribution (i.e., the ones affected) are concentrated very near the threshold. Thus, even slightly reducing their liability can move a large proportion of affected individuals below the disease threshold.

The argument that large reductions in relative risk can be achieved even with small reductions in the PRS is fundamental to our paper. As such, we regret not fully explaining these arguments in the original manuscript. In the revised manuscript, we have substantially expanded on the intuition behind this result.

2. It appears to be counter-intuitive that the curves of the "high-risk exclusion" strategy would peak instead of being monotone (e.g. the upper-left panel in figure 2). If the exclusion percentile (i.e., x-axis) keeps increasing, shouldn't this approach behave similarly compared to the "low-risk prioritization" strategy? One would think that a stringent percentile threshold in "high-risk exclusion" (excluding most embryos) would ensure that only embryos with very low risks are planted, which is what the "low-risk prioritization" strategy tries to achieve. So why doesn't the purple curve in that upper-left panel keep going up until reaching ~80% which is suggested by the lower-left panel in Figure 2?

We agree that this result may initially seem counter-intuitive, and, in the revised manuscript, we added an explanation. The fact that the relative risk reduction is not monotonically increasing under the high-risk exclusion (HRE) strategy is not an error. To see this, recall that we defined the HRE strategy such that when all embryos are high risk, an embryo is selected at random. When the exclusion threshold is set to the top few percentiles of the PRS (e.g., top 2% or top 10% of the PRS), it is almost always the case that at least one embryo is not high risk and thus can be implanted (say, for n=5 embryos). However, when loosening the threshold (e.g., designating embryos at the top 50% of the PRS distribution as high risk), it can become common that all embryos are high risk. In such cases, selection of the embryo for implantation is random. At the extreme case, when the top 100% of embryos (i.e., all embryos) are designated as high risk, an embryo is selected at random at all times, and the relative risk reduction reduces to zero.

We can also consider an alternative HRE strategy, a mixture of the “pure” HRE strategy and the lowest-risk prioritization (LRP) strategy. In this alternative, if all embryos are high risk, the embryo with the lowest risk score is selected. Here, the relative risk reduction is expected to increase when extending the definition of high-risk embryos to include more PRS percentiles. At the extreme case, when all embryos are designated as high risk, this strategy reduces to the LRP strategy. While we do not have an analytical solution for the relative risk reduction under this alternative strategy, it is easy to simulate. In Figure 2 —figure supplement 2 in the revised manuscript , we show simulation results demonstrating the behavior of the various strategies, in agreement with the qualitative arguments we made above.

We expect that the HRE strategy will be applied in practice with very stringent thresholds for defining high-risk embryos (e.g., top 2% or top 10% PRS percentile). This is because these thresholds correspond to current practices in clinical genetics, and because such a choice may be perceived as circumventing ethical issues associated with selecting the “best” embryo. As we mentioned above, in this scenario, there is almost always at least one non-high risk embryo that could be implanted. And thus, the overall risk reduction is only little affected by the decision on how an embryo is selected when all embryos are high-risk. Indeed, as can be seen in Figure 2—figure supplement 2, as long as high-risk embryos are defined as having a PRS within the top ≈20% (or more stringently), both the “pure” HRE and the “mixed” strategy result in very similar risk reductions.

3. The authors should comment on the following points. First, partitioning the PRS "s_i_" into two orthogonal components: polygenic transmission disequilibrium "x_i_" and parental average score "c" seems intuitive. However, shouldn't parental genotypes predict the distribution of x_i_ ? Suppose both parents are homozygous at a SNP and c = (2+2)/2 = 2. Then, s_i_ would be 2 without any variability, which means x_i_ is always 0. Even in a polygenic setting, a very high value of c would suggest higher homozygosity which should lead to lower variance of x_i_. However, the setting in this paper leads to a constant variance of x_i_ which isn't affected by c. Is something missing here?

The reviewer’s example is of course correct, and it is true that when considering a single locus, there may be no variance in the genotype of the child. However, our model focuses on polygenic diseases, which are typically governed by variation in thousands of loci. For example, (O'Connor et al., 2019) have recently estimated the number of independent loci associated with 33 phenotypes (Table 1 therein). With the exception of red hair and sunburn, the number of loci was at least ≈350, and exceeded ≈10,000 for cognitive and psychiatric phenotypes.

Consider now the following simple argument. Say the number of loci is m=500 (which is at the lower end of polygenicity across phenotypes), and, for the purpose of illustration, assume all variants have the same effect size and that they all have allele frequency 50%. The risk score is then simply the count of risk alleles, out of 2m=1000 potential copies. This binomial variable can be replaced with a normal with mean 2m⋅0.5=500 and a standard deviation 2m⋅0.5⋅0.5≈16. In this setting, even an individual at the top 0.1% quantile (top 1/1000) will have only ≈550 risk alleles (and, thus, ≈450 reference alleles). This is far from a situation where all parental loci are homozygous for the risk allele, and leaves plenty of room for a substantial difference between the children (the embryos). It is thus sensible to assume that the variance in the embryos’ score around the parental average is independent of that average.

In the revised manuscript, we also provide empirical support to the independence of the variance on the parental scores, based on real genomic data from case-control studies of schizophrenia and Crohn’s disease. See a detailed description in our response to comment 13.

We discuss these points in the thoroughly revised Discussion section, where we also emphasize that our results hold only for polygenic diseases.

4. The authors state that they have not explored the age dependence on the number of embryos? How significant would this omission be? What is the age distribution of parents who would typically undertake such polygenic embryo selection?

We did not model the age dependence on the number of embryos: as polygenic embryo selection is a new technology, we do not know what will be the age distribution of its future users. [As a side note, we have recently initiated a project, jointly with colleagues, to explore attitudes of IVF couples towards polygenic embryo screening. We expect that the results will also provide information on the expected age distribution.] To avoid this uncertainty, we have specified all of our results in terms of the number of available viable embryos. This would allow readers to easily determine the expected outcomes given the number of embryos available in any age group of interest.

Nevertheless, in light of this comment, as well as comment (9) below, in the revised manuscript we expand on the implications of assuming that a given number of embryos is available. Specifically, we emphasize that embryos included within the embryo count must have the potential to lead to *live birth*. We further emphasize that this is a more stringent criterion than developing normally in vitro. In fact, many IVF cycles lead to no live birth at all. For example, the live birth rate is 32% for women under 40, and decreases to 12% for women aged 40-42 and 4% for women over 42 (Smith et al., 2015). However, birth rates are higher for younger women undergoing elective oocyte cryopreservation. We also explicitly point out in the revised manuscript that even with the lowest-risk prioritization strategy, the risk reduction decreases very rapidly when the number of available embryos decrease below n≈5.

5. For the high risk exclusion strategy, the authors state that if all embryos are of high risk, then a random embryo would be implanted. Would a decision of not implanting any embryo be a reasonable alternative? If this is indeed a reasonable alternative, would this change the authors' conclusion in any way?

We note that this question relates to comment (2) above. In our reply to that comment, we described an alternative strategy, under which, when all embryos are high risk, the embryo with the lowest PRS is selected. The results (Figure 2 – Figure Supplement 2 in the revised manuscript) show that the risk reduction under this sub-strategy is intermediate between the “pure” high-risk exclusion (HRE) and the lowest-risk prioritization (LRP) strategies. However, as long as the high-risk embryos are narrowly defined as having a very high PRS (e.g., top 2% or top 10% of the PRS distribution), the risk reduction under this strategy is about the same that of the usual HRE strategy.

We agree that another possibility is not to implant any embryo in case all embryos are high risk. However, at least in the case of n=5 embryos, a realistic scenario in which all embryos are high-risk is not very likely. For example, consider parents who both have an average PRS (i.e., zero), and assume that an embryo is classified as high-risk only if its PRS is at the top 2%. The PRS of each child is the parental average (zero) + a random component, which has half the population variance of the PRS. The probability of a child to have PRS in the top 2% is 0.002 (pnorm(qnorm(0.98),0,sqrt(0.5),lower.tail = F)). As the scores of the embryos are independent conditional on the parental scores, the probability of all five embryos to have PRS in the top 2% is only 2·10^-14^. If the parental average is at the 90% PRS percentile, the probability of each embryo to have PRS in the top 2% is 0.14 (pnorm(qnorm(0.98),qnorm(0.9),sqrt(0.5),lower.tail = F)), and the probability of all five to be high-risk is 0.00005. Finally, if the parental average is at the 98% PRS percentile, the probability of each embryo to have a PRS in the top 2% is (trivially) 0.5. However, even so, the probability of all five embryos to be high-risk is only 0.03.

Regarding the reviewers’ highlighted scenario of no implantation at all, we are not certain how to define the risk reduction in such a case. This is because, if no child will be born, there is no way to assess the risk of disease in the child. Relatedly, we note that in a case report of the first clinical application of polygenic embryo screening (Treff et al., 2019a), the couple has elected not to implant any embryo, even though 3/5 embryos were not high risk. No further details were provided for the reasons behind this decision, but this case nevertheless demonstrates that no implantation is another possible outcome of polygenic embryo screening. We revised the manuscript accordingly to reflect this.

6. The authors should clarify the impact of error in the per-embryo estimation of x_s_ (e.g. from errors in imputation of genotypes if that's used or errors in the β_hat estimates that form the PRS)? I assume in these cases it results in an effective lowering of r^2^_ps_?

Sequencing the DNA of an embryo based on a single cell or just a few cells is indeed more error prone (mostly due to allele drop-out) than sequencing bulk DNA from a tissue of a living person. The problem is amplified if the sequencing depth per embryo is low. Sequencing errors could then lead to noise in the polygenic score assigned to each embryo. However, in practice, the genomes of embryos can be reconstructed with very high accuracy. The underlying reason is that in preimplantation genetic testing, it is always possible to sequence the prospective parents, based on bulk DNA, to very high quality. Usually, the sequencing of the parents is accompanied by sequencing of one of their first degree relatives, which further allows accurate phasing the parental haplotypes. Then, to reconstruct the genome of an embryo, all that is needed is to infer is the locations of crossovers, which is usually not difficult using hidden Markov models or similar methods. For example, see papers by us and others: (Backenroth et al., 2019; Handyside et al., 2010; Kumar et al., 2015; Natesan et al., 2014; Treff et al., 2019b; Xiong et al., 2019; Xu et al., 2015; Yan et al., 2015; Zamani Esteki et al., 2015). Once crossovers have been identified, their locations imply which of the paternal and maternal haplotypes have been transmitted to each embryo at each locus. Given that the parental haplotypes are known with high accuracy, the genome of the embryo is thus fully specified. While *de-novo* mutations cannot be detected using this class of methods, such mutations are usually not included in polygenic scores.

Even if sequencing errors do occur, their effect can be readily incorporated within our modeling framework. Specifically, suppose that r02 is the proportion of variance explained by a perfectly genotyped score, and that rimpute2 is the squared correlation between the true score and the imputed score of an embryo. Based on properties of simple linear regression, rps2=r02⋅rimpute2, where rps2 is the variance explained by the (observed) score, i.e., is the quantity used in our models. We did not explicitly model rimpute2 in this paper, but it would be easy to estimate it by any lab developing PGT methods. Either way, all of our results are presented as a function of rps2, which would allow readers to factor in the effect of imperfectly genotyped scores. In the revised manuscript, we added a few sentences in the *Materials and methods* to discuss this point.

7. While the statistical framework in this paper can be used to compare the utility of different strategies, the same analyses can also quantify the potential risks of embryo screening. Although the authors were open about the limitations of the study, no empirical analyses were performed to demonstrate potential risks. This is a missed opportunity. Can the authors expand this somewhat?

As suggested by the reviewer, we have now added consideration of risks of embryo screening to the Discussion section of the revised manuscript. Polygenic embryo screening and prioritization involves three major types of risk. The first is direct risk to the mother (Gelbaya, 2010; Dayan et al., 2019) and child (Hart and Norman, 2013; Luke et al., 2018) due to the IVF procedure. However, regarding the risk of IVF, (Gelbaya, 2010) wrote “most of these risks, however, are related to the woman's subfertility status and/or increased incidence of multiple pregnancy”. Either way, IVF is very widely applied (e.g., https://www.sciencedaily.com/releases/2018/07/180703084127.htm) and any long-term health issues associated with IVF are not specific to our study. A second form of risk is the risk to the embryo due to the biopsy required for genotyping; for which the evidence is mixed. This evidence is briefly reviewed in the revised manuscript. However, it is possible that newly developed techniques may allow in the future to genotype an embryo non-invasively based on DNA present in spent culture medium, although the accuracy of these methods is still being debated (Leaver and Wells, 2020).

The third type of risk associated with polygenic embryo screening is genetic. This can happen when prioritizing embryos for implantation based on the risk for one disease, and thereby unintentionally increasing the risk of a second disease (which was not screened). This will be the case if the second disease is genetically negatively correlated with the first. We address this point in the response to the next comment.

8. While it is understandable to make some simplifying assumptions, It is critical to demonstrate the risk which is a main concern of human embryo screening in the field. While it does not appear that the authors have over-stated anything in the original submission, given the sensitivity of this topic, avoiding quantitative discussions on risks may lead to over-optimistic news headlines that only focus on the potential utility and even reckless and potentially harmful practices. With that said, it seems straightforward to generalize the simulation framework in this paper to explore a two-trait setting. Suppose two diseases are genetically correlated (with correlation = -0.1, -0.3, and -0.5, for example). Then, would the "low-risk prioritization approach" on one disease lead to drastically increased risks for the second disease? It wouldn't be fair to ask the authors to find a solution to this problem given the scope of the study. But a quantitative demonstration of risks will improve this paper.

We agree with the reviewers that this point is important to quantify. As we describe in the new section 7 of the *Materials and methods*, we developed a simple model for the distribution of the scores for two diseases. Using our model, we could simulate scores of two correlated diseases in batches of n embryos. While a full analysis of screening for multiple diseases is left for future work, our simulation framework allowed us to investigate the potential harmful effects of prioritizing embryos for one disease, in case that disease is anti-correlated with another disease. We present numerical results in Figure 5. We considered genetic correlations between diseases taking the values ρ=(−0.05,−0.1,−0.15,−0.2,−0.3). While the reviewers recommended considering genetic correlations as negative as ρ=−0.5, the most negative correlation we are aware of is -0.3, occurring between ulcerative colitis and chronic kidney disease (Zheng et al., 2017). In general, negative correlations between diseases are uncommon, and when they occur, typical values are about -0.1.

Figure 5 shows the risk *reduction* for the *target* disease and the risk *increase* for the *correlated* disease, across different values of ρ and for three values of the prevalence K (panels (A)-(C); assumed equal for the two diseases). In all panels, we set rps2=0.1 for both diseases. The relative risk reduction for the target disease is, as expected, always higher in absolute value than the risk *increase* for the correlated disease. For typical values of ρ=−0.1 and n=5, the relative *increase* in risk of the correlated disease is relatively small, at ≈6% for K≤0.05 and ≈3.5% for K=0.2. However, for strong negative correlations (ρ=−0.3) the increase in risk can reach 22%, 16%, or 11% for K=0.01, 0.05 and 0.2, respectively.

Therefore, the results demonstrate that care must be taken in the unique setting when the target disease is *strongly* negatively correlated with another disease. In the revised manuscript, we expand on this important observation. Specifically, we added a section to the *Materials and methods* describing the simulation framework, we added a section in the main text describing the numerical results.

9. What is the current success rate if only one embryo is planted? If multiple embryos (out of 5) are planted to ensure a reasonable success rate in current practices, would it be fair to say that only a "high-risk exclusion"-type strategy that only excludes embryos with higher risk is realistic? Here, the exclusion criterion isn't necessarily based on a population quantile but could be based on PRS ranking among embryos. Some discussions will be helpful.

It is possible to generalize both the “high-risk exclusion” (HRE) and the “lowest-risk prioritization” (LRP) strategies to the case of transfer of multiple embryos. Under HRE, any two random embryos can be selected for implantation, as long as both have PRS under the high-risk cutoff. (And if only one or no such embryo exists, up to two embryos can be selected randomly out of all embryos.) Under LRP, the two embryos with the lowest PRS would be transferred. In terms of the statistical model, we would need to compute the joint distribution of the scores of the two selected embryos. Then, ignoring here the possibility of twin birth, the disease risk could be computed assuming an equal probability (0.5) for each of those two embryos to be born. However, the joint distribution of the scores of the selected embryos is difficult to derive. Alternatively, it is straightforward to simulate the transfer of multiple embryos: once two embryos are selected (based on either strategy, as explained above), one of them is randomly selected, and the disease risk can be calculated as for a single embryo transfer. This can be easily implemented based on our existing code. We expect the relative risk reduction to somewhat decrease (under both strategies), due to the possibility of an implantation of an embryo with a higher PRS compared to single embryo transfer.

In the revised manuscript, we mention the possibility of multiple embryo transfer and how selection strategies would change accordingly (*Materials and methods*). However, we opted not to add analyses and results, for the following two reasons. First, following the revision, the paper is already very dense. Second, IVF clinics around the world are moving away from multiple embryo transfer (De Geyter et al., 2020), due to the health risks associated with twin or higher order pregnancies.

10. The authors state that one concern about the generalisability of polygenic embryo selection is the extension to non-European populations. Outside of the European populations, is it true that none of the other ethnic groups have high quality polygenic risk score data?

The lower accuracy of polygenic scores in non-European populations will decrease the risk reduction due to the lower variance explained by the score. However, the theoretical risk reduction that we derived in our paper is always presented as a function of rps2, the proportion of variance (in liability) explained by the score. Thus, variable PRS accuracy across populations does not present a conceptual problem, because all that is needed is to substitute the value of rps2 relevant to the target population. Several papers, including a few very recent ones, have quantified the variance explained by polygenic scores across populations and traits (Duncan et al., 2019; Fahed et al., 2020; Kember et al., 2019; Lehmann et al., 2021; Majara et al., 2021; Martin et al., 2019; Privé et al., 2021).

However, it is also important to note that prediction accuracy in non-European populations is far from being null. In fact, some papers have demonstrated high prediction accuracies in East Asians (Duncan et al., 2019) or South Asians (Martin et al., 2019). Further, the predictive accuracy of polygenic scores in non-Europeans is expected to rapidly improve in the near future, due to the following three major developments.

The first is the establishment of very large biobanks covering individuals with ancestries other than North-Western European. Examples include Biobank Japan, China Kadoorie biobank, Born in Bradford and East London Genes and Health studies in the UK (which both include individuals of predominantly South Asian ancestry), All of US and the Million Veteran Program in the US (which both have a substantial proportion of individuals of minority ancestries), and one may also count FinnGen and The Estonia Biobank.

The second development is the rapid advance in methods that can improve the accuracy of PRSs across ethnicities, e.g., using data from large European GWASs coupled with smaller studies in the target population (Bitarello and Mathieson, 2020; Cavazos and Witte, 2021; Coram et al., 2017; Grinde et al., 2019; Liang et al., 2020; Marnetto et al., 2020; Marquez-Luna et al., 2017; Ruan et al., 2021).

Third, there is a growing recognition that causal effect sizes are correlated (even if imperfectly) between populations. Consequently, the low predictive performance of PRSs in non-Europeans results largely from differences in linkage disequilibrium and allele frequencies, and less so due to differences in the identity and the effect sizes of the underlying causal variants (Bitarello and Mathieson, 2020; Cavazos and Witte, 2021; Chen et al., 2020; Ishigaki et al., 2020; Lam et al., 2019; Shi et al., 2020; Shi et al., 2021; Wang et al., 2020; Wang et al., 2021; Wojcik et al., 2019). Thus, progress in methods for fine-mapping (identifying causal variants) is promising for generating polygenic scores that remain accurate across populations even when they are derived from GWASs in Europeans (Amariuta et al., 2020; Weissbrod et al., 2021).

In summary, we provide all of our results as a function of rps2, allowing users to substitute values corresponding to their target population. Further, even though current predictive accuracy is lower in non-European populations, accuracy is rapidly improving due to the accumulation of genetic data and an improvement in methods for transferring scores across populations. We included a brief summary of this argument in the revised Discussion.

11. The usage of an extended listing of assumptions deep in the Material and Methods was slightly unorthodox to me and I wonder if it would be more transparent and useful for the average reader's consumptions to include these in the discussion itself. While the presented results are helpful, it's important that stakeholders digesting these numbers understand how many unknowns there are to still understand in this area.

We agree with this comment. We initially aimed at a short report, and therefore attempted to keep the main text short. However, given that the manuscript has substantially expanded, this is no longer needed. Accordingly, we added to the Discussion an overview of the assumptions and limitations. We retained and expanded the Limitations section in the *Materials and methods* to cover the more technical model assumptions.

12. Beyond the average risk reduction, would it be helpful to understand the variance in outcomes, e.g., which strategies have high variance in outcome? Can the authors speak to that dimension even qualitatively?

The issue of variance in outcomes is important, and in the revised manuscript we have substantially expanded its treatment.

In the original manuscript, we defined the relative risk reduction (RRR) as follows:(1)RRR=1−Ps(disease)Pr(disease)

In Equation (1), Ps(disease) is the probability of a random couple to have an affected child, after *s*electing an embryo based on its PRS (according to any given strategy). Pr(disease) is the same probability for a *r*andom embryo (or equivalently for a natural conception). From a different angle, Ps(disease) is the prevalence of the disease in a population in which all embryos are selected based on their PRS, and Pr(disease) is the prevalence in a population in which all embryos are selected at random. The RRR is the (complement of the) ratio between these two average risks.

Each of the probabilities in Equation (1) is an average over *three sources of variability*:

1. The genetic composition of the parents.

2. The scores of the embryos generated in a given IVF cycle.

3. The non-score genetic factors and the environmental factors to be experienced by the selected embryo.

In the revised manuscript, we propose that the averaging over (1) could be removed, thereby providing a *per-couple* estimate of the relative risk reduction, or pcRRR. [We also explain, though we don’t numerically analyze, how to remove the averaging over both (1) and (2), thereby providing a risk reduction estimate *per-batch* of embryos belonging to a given couple.]

To understand how to compute the *per-couple* relative risk reduction, pcRRR, we note that the only relevant information about the PRS that distinguishes one pair of prospective parents from another is c, the average of the paternal and maternal PRSs. In other words, all couples with the same value of c will have the same relative risk reduction. We have already computed the disease risk conditional on c, under all strategies (*lowest-risk prioritization* (LRP), *high-risk exclusion* (HRE), and random selection), in the original manuscript. Thus, the pcRRR(c) is simply defined as:(2)pcRRR(c)=1−Ps(disease|c)Pr(disease|c)

In Equation (2), Ps(disease |c) is the probability of the implanted embryo (selected by a given strategy) to be affected, averaged over all possible batches of n embryos that could have been generated for a couple with an average PRS c. The corresponding term for random selection, Pr(disease |c), is similarly defined.

Thus, couples with different average parental scores will have different risk reductions. We investigated the variability across couples by computing and plotting the distribution of pcRRR across couples (see the revised Materials and methods for details on the computation). The results (for the LRP strategy) are shown in Figure 4 (panels A-C), demonstrating that the pcRRR is relatively narrowly distributed around its mean, for all values of the prevalence (K) considered. The distribution becomes somewhat wider (and left-tailed) for the most extreme rps2 (0.3). The narrow distribution results from the fact that the pcRRR depends only weakly on the average parental PRS, as can be seen in panels D-F. Exceptions are observed only for rps2=0.3 and only for the most extreme parental scores.

13. Perhaps one set of simulations with a finite and varying number of loci would be useful to guide intuition on when the infinitesimal model is appropriate. From experiences in other examples, the infinitesimal can already provide useful predictions for what might at first seem like small numbers of loci and if that holds in this case it might be useful for readers to understand who might otherwise be too quickly dismissive.

Indeed, our theory is based on the polygenic model of disease, which posits that the underlying (genetic and non-genetic) components of the liability are influenced by numerous factors. Thus, our model did not explicitly consider the number of loci, but rather assumed that all scores and other components are normally distributed. While the polygenic liability threshold model is popular in the statistical genetics literature (see the *Materials and methods* Section 1), it is also important to verify that our predictions hold when applied to real data.

Clearly, no real genomic/phenotypic data exist on embryos that would correspond to our setting, nor could such data be ethically/practically generated. We could nevertheless test our model using a “hybrid” approach, in which we simulated the genomes of embryos based on real genomes from case-control studies. This approach is similar to how we have previously studied embryo screening for continuous traits (Karavani et al., 2019).

Our approach is as follows. We consider separately two diseases: schizophrenia and Crohn’s disease. For each disease, we use genomes of unrelated individuals from case control studies. For schizophrenia, we use ≈900 cases and ≈1600 controls (Lencz et al., 2013). For Crohn’s, we use ≈150 cases and ≈100 controls. We then generate “virtual couples” by randomly mating pairs of individuals, regardless of sex, but accounting for the disease prevalence. For each couple, we simulate the genomes of n hypothetical embryos, based on the laws of Mendelian inheritance and by randomly placing crossovers according to genetic map distances. In parallel, we use the “parental” genomes to learn a logistic regression model that predicts the disease risk given a PRS. We then compute the PRS of each simulated embryo, and predict the risk of disease of that embryo. Finally, we compare the risk of disease between a population in which one embryo per couple is selected at random, vs a population in which one embryo is selected based on PRS. We compared the empirical relative risk reduction to that predicted by the theory, after substituting the prevalences of these diseases and the empirically observed variance explained by the PRSs. For more details, see the revised *Materials and methods*.

The results of the analysis are presented in Figure 6, showing very good agreement with the theory for both schizophrenia (panels (A) and (B)) and Crohn’s (panels (C) and (D)). The analytical predictions closely match the empirical risk reductions for both the HRE and LRP strategies, though with a slight overestimation of the risk reduction under LRP. We also investigated intermediate predictions made by the model. Specifically, one prediction is that the variance of the PRSs across embryos (for a given couple) should not depend on the average PRS of the parents. A second prediction is that the variance across embryos should be half the “population-level” variance of the PRS. In Figure 6 —figure supplement 1, we use our schizophrenia and Crohn’s disease cohorts to evaluate these predictions. We find overall good agreement between theoretical expectations and empirical results for both predictions. The main deviation was an uptick of the variance at very low parental PRSs for schizophrenia. However, individuals with low PRS are less relevant from the point of view of screening for eliminating disease. In total, these encouraging results, across two very severe diseases with different genetic architectures, provide strong support for the relevance of our statistical model and assumptions.

In the revised manuscript, we added an entire new section presenting these results.

14. For clarity, please use a different variable besides in 'e' in sections where 'e' includes environment plus unmeasured genetic effects. Perhaps 'u' for "un-measured" effects?

We understand the concern of the reviewers. To protect from ambiguity, we have made sure to explicitly clarify that e is used to describe any factor (genetic or environmental) beyond the measureable PRS. We used e because it represents the residual of the regression of the trait (liability in our case) on the PRS, and as such, it is commonly denoted by e. Note that we have reserved the Greek letter ϵ to describe purely environmental factors (see in particular Section 6, where this notation is used extensively). We believe that at this point, when the *Materials and methods* extend to ≈30 pages, changing a fundamental notation may lead to downstream errors and may inadvertently increase confusion.

15. Equation 16, e_i_* was not explicitly defined, even if it's inferable from context it would be nice to define.

We corrected this omission.

References

Amariuta, T., Ishigaki, K., Sugishita, H., Ohta, T., Matsuda, K., Murakami, Y., Price, A.L., Kawakami, E., Terao, C., and Raychaudhuri, S. (2020). In silico integration of thousands of epigenetic datasets into 707 cell type regulatory annotations improves the trans-ethnic portability of polygenic risk scores. bioRxiv, 2020.2002.2021.959510.

Backenroth, D., Zahdeh, F., Kling, Y., Peretz, A., Rosen, T., Kort, D., Zeligson, S., Dror, T., Kirshberg, S., Burak, E.*, et al.* (2019). Haploseek: a 24-hour all-in-one method for preimplantation genetic diagnosis (PGD) of monogenic disease and aneuploidy. Genetics in medicine : official journal of the American College of Medical Genetics *21*, 1390-1399.

Bitarello, B.D., and Mathieson, I. (2020). Polygenic Scores for Height in Admixed Populations. G3 (Bethesda) *10*, 4027-4036.

Cavazos, T.B., and Witte, J.S. (2021). Inclusion of variants discovered from diverse populations improves polygenic risk score transferability. HGG Adv *2*.

Chen, M.H., Raffield, L.M., Mousas, A., Sakaue, S., Huffman, J.E., Moscati, A., Trivedi, B., Jiang, T., Akbari, P., Vuckovic, D.*, et al.* (2020). Trans-ethnic and Ancestry-Specific Blood-Cell Genetics in 746,667 Individuals from 5 Global Populations. Cell *182*, 1198-1213 e1114.

Cimadomo, D., Capalbo, A., Ubaldi, F.M., Scarica, C., Palagiano, A., Canipari, R., and Rienzi, L. (2016). The Impact of Biopsy on Human Embryo Developmental Potential during Preimplantation Genetic Diagnosis. Biomed Res Int *2016*, 7193075.

Coram, M.A., Fang, H., Candille, S.I., Assimes, T.L., and Tang, H. (2017). Leveraging Multi-ethnic Evidence for Risk Assessment of Quantitative Traits in Minority Populations. American journal of human genetics *101*, 638.

Dayan, N., Joseph, K. S., Fell, D. B., Laskin, C. A., Basso, O., Park, A. L., Luo, J., Guan, J., and Ray, J. G. (2019). Infertility treatment and risk of severe maternal morbidity: a propensity score-matched cohort study. CMAJ: Canadian Medical Association Journal = Journal de l’Association Medicale Canadienne, 191(5), E118–E127.

De Geyter, C., Wyns, C., Calhaz-Jorge, C., de Mouzon, J., Ferraretti, A.P., Kupka, M., Nyboe Andersen, A., Nygren, K.G., and Goossens, V. (2020). 20 years of the European IVF-monitoring Consortium registry: what have we learned? A comparison with registries from two other regions. Hum Reprod *35*, 2832-2849.

Duncan, L., Shen, H., Gelaye, B., Meijsen, J., Ressler, K., Feldman, M., Peterson, R., and Domingue, B. (2019). Analysis of polygenic risk score usage and performance in diverse human populations. Nat Commun *10*, 3328.

Fahed, A.C., Aragam, K.G., Hindy, G., Chen, Y.I., Chaudhary, K., Dobbyn, A., Krumholz, H.M., Sheu, W.H.H., Rich, S.S., Rotter, J.I.*, et al.* (2020). Transethnic Transferability of a Genome-wide Polygenic Score for Coronary Artery Disease. Circ Genom Precis Med.

Gelbaya, T.A. (2010). Short and long-term risks to women who conceive through in vitro fertilization. Hum Fertil (Camb) *13*, 19-27.

Grinde, K.E., Qi, Q., Thornton, T.A., Liu, S., Shadyab, A.H., Chan, K.H.K., Reiner, A.P., and Sofer, T. (2019). Generalizing polygenic risk scores from Europeans to Hispanics/Latinos. Genetic epidemiology *43*, 50-62.

Handyside, A.H., Harton, G.L., Mariani, B., Thornhill, A.R., Affara, N., Shaw, M.A., and Griffin, D.K. (2010). Karyomapping: a universal method for genome wide analysis of genetic disease based on mapping crossovers between parental haplotypes. J Med Genet *47*, 651-658.

Hart, R., and Norman, R.J. (2013). The longer-term health outcomes for children born as a result of IVF treatment: Part I--General health outcomes. Hum Reprod Update *19*, 232-243.

Ishigaki, K., Akiyama, M., Kanai, M., Takahashi, A., Kawakami, E., Sugishita, H., Sakaue, S., Matoba, N., Low, S.K., Okada, Y.*, et al.* (2020). Large-scale genome-wide association study in a Japanese population identifies novel susceptibility loci across different diseases. Nature genetics *52*, 669-679.

Karavani, E., Zuk, O., Zeevi, D., Barzilai, N., Stefanis, N.C., Hatzimanolis, A., Smyrnis, N., Avramopoulos, D., Kruglyak, L., Atzmon, G.*, et al.* (2019). Screening Human Embryos for Polygenic Traits Has Limited Utility. Cell *179*, 1424-1435 e1428.

Kember, R.L., Verma, A., Verma, S., Lucas, A., Judy, R., Chen, J., Damrauer, S., Rader, D.J., and Ritchie, M.D. (2019). Polygenic Risk Scores for Cardio-renal-metabolic Diseases in the Penn Medicine Biobank. bioRxiv, 759381.

Kumar, A., Ryan, A., Kitzman, J.O., Wemmer, N., Snyder, M.W., Sigurjonsson, S., Lee, C., Banjevic, M., Zarutskie, P.W., Lewis, A.P.*, et al.* (2015). Whole genome prediction for preimplantation genetic diagnosis. Genome Med *7*, 35.

Lam, M., Chen, C.Y., Li, Z., Martin, A.R., Bryois, J., Ma, X., Gaspar, H., Ikeda, M., Benyamin, B., Brown, B.C.*, et al.* (2019). Comparative genetic architectures of schizophrenia in East Asian and European populations. Nature genetics *51*, 1670-1678.

Leaver, M., and Wells, D. (2020). Non-invasive preimplantation genetic testing (niPGT): the next revolution in reproductive genetics? Hum Reprod Update *26*, 16-42.

Lee, S.H., Goddard, M.E., Wray, N.R., and Visscher, P.M. (2012). A better coefficient of determination for genetic profile analysis. Genetic epidemiology *36*, 214-224.

Lehmann, B.C.L., Mackintosh, M., McVean, G., and Holmes, C.C. (2021). High trait variability in optimal polygenic prediction strategy within multiple-ancestry cohorts. bioRxiv, 2021.2001.2015.426781.

Lencz, T., Guha, S., Liu, C., Rosenfeld, J., Mukherjee, S., Derosse, P., John, M., Cheng, L., Zhang, C., Badner, J.A.*, et al.* (2013). Genome-wide association study implicates NDST3 in schizophrenia and bipolar disorder. Nat Commun *4*, 2739.

Li, M., Kort, J., and Baker, V.L. (2021). Embryo biopsy and perinatal outcomes of singleton pregnancies: an analysis of 16,246 frozen embryo transfer cycles reported in the Society for Assisted Reproductive Technology Clinical Outcomes Reporting System. Am J Obstet Gynecol *224*, 500 e501-500 e518.

Liang, Y., Pividori, M., Manichaikul, A., Palmer, A.A., Cox, N.J., Wheeler, H., and Im, H.K. (2020). Polygenic transcriptome risk scores improve portability of polygenic risk scores across ancestries. bioRxiv, 2020.2011.2012.373647.

Luke, B. (2017). Pregnancy and birth outcomes in couples with infertility with and without assisted reproductive technology: with an emphasis on US population-based studies. *American Journal of Obstetrics and Gynecology*, *217*(3), 270–281.

Majara, L., Kalungi, A., Koen, N., Zar, H., Stein, D.J., Kinyanda, E., Atkinson, E.G., and Martin, A.R. (2021). Low generalizability of polygenic scores in African populations due to genetic and environmental diversity. bioRxiv, 2021.2001.2012.426453.

Marnetto, D., Parna, K., Lall, K., Molinaro, L., Montinaro, F., Haller, T., Metspalu, M., Magi, R., Fischer, K., and Pagani, L. (2020). Ancestry deconvolution and partial polygenic score can improve susceptibility predictions in recently admixed individuals. Nat Commun *11*, 1628.

Marquez-Luna, C., Loh, P.R., South Asian Type 2 Diabetes, C., Consortium, S.T.D., and Price, A.L. (2017). Multiethnic polygenic risk scores improve risk prediction in diverse populations. Genetic epidemiology *41*, 811-823.

Martin, A.R., Kanai, M., Kamatani, Y., Okada, Y., Neale, B.M., and Daly, M.J. (2019). Clinical use of current polygenic risk scores may exacerbate health disparities. Nature genetics *51*, 584-591.

Natesan, S.A., Bladon, A.J., Coskun, S., Qubbaj, W., Prates, R., Munne, S., Coonen, E., Dreesen, J.C., Stevens, S.J., Paulussen, A.D.*, et al.* (2014). Genome-wide karyomapping accurately identifies the inheritance of single-gene defects in human preimplantation embryos in vitro. Genetics in medicine : official journal of the American College of Medical Genetics *16*, 838-845.

Natsuaki, M.N., and Dimler, L.M. (2018). Pregnancy and child developmental outcomes after preimplantation genetic screening: a meta-analytic and systematic review. World J Pediatr *14*, 555-569.

O'Connor, L.J., Schoech, A.P., Hormozdiari, F., Gazal, S., Patterson, N., and Price, A.L. (2019). Extreme Polygenicity of Complex Traits Is Explained by Negative Selection. American journal of human genetics *105*, 456-476.

Pinborg, A. (2019). Short- and long-term outcomes in children born after assisted reproductive technology. BJOG *126*, 145-148.

Privé, F., Aschard, H., Folkersen, L., Hoggart, C., O’Reilly, P.F., and Vilhjálmsson, B.J. (2021). High-resolution portability of 240 polygenic scores when derived and applied in the same cohort. medRxiv, 2021.2002.2005.21251061.

Riestenberg, C.K., Mok, T., Ong, J.R., Platt, L.D., Han, C.S., and Quinn, M.M. (2021). Sonographic abnormalities in pregnancies conceived following IVF with and without preimplantation genetic testing for aneuploidy (PGT-A). J Assist Reprod Genet.

Ruan, Y., Anne Feng, Y.-C., Chen, C.-Y., Lam, M., Sawa, A., Martin, A.R., Qin, S., Huang, H., and Ge, T. (2021). Improving Polygenic Prediction in Ancestrally Diverse Populations. medRxiv, 2020.2012.2027.20248738.

Savage, J.E., Jansen, P.R., Stringer, S., Watanabe, K., Bryois, J., de Leeuw, C.A., Nagel, M., Awasthi, S., Barr, P.B., Coleman, J.R.I.*, et al.* (2018). Genome-wide association meta-analysis in 269,867 individuals identifies new genetic and functional links to intelligence. Nature genetics *50*, 912-919.

Scott, R.T., Jr., Upham, K.M., Forman, E.J., Zhao, T., and Treff, N.R. (2013). Cleavage-stage biopsy significantly impairs human embryonic implantation potential while blastocyst biopsy does not: a randomized and paired clinical trial. Fertil Steril *100*, 624-630.

Shi, H., Burch, K.S., Johnson, R., Freund, M.K., Kichaev, G., Mancuso, N., Manuel, A.M., Dong, N., and Pasaniuc, B. (2020). Localizing Components of Shared Transethnic Genetic Architecture of Complex Traits from GWAS Summary Data. American journal of human genetics *106*, 805-817.

Shi, H., Gazal, S., Kanai, M., Koch, E.M., Schoech, A.P., Siewert, K.M., Kim, S.S., Luo, Y., Amariuta, T., Huang, H.*, et al.* (2021). Population-specific causal disease effect sizes in functionally important regions impacted by selection. Nat Commun *12*, 1098.

Smith, A., Tilling, K., Nelson, S.M., and Lawlor, D.A. (2015). Live-Birth Rate Associated With Repeat in vitro Fertilization Treatment Cycles. JAMA *314*, 2654-2662.

Tiegs, A.W., Tao, X., Zhan, Y., Whitehead, C., Kim, J., Hanson, B., Osman, E., Kim, T.J., Patounakis, G., Gutmann, J.*, et al.* (2020). A multicenter, prospective, blinded, nonselection study evaluating the predictive value of an aneuploid diagnosis using a targeted next-generation sequencing-based preimplantation genetic testing for aneuploidy assay and impact of biopsy. Fertil Steril.

Treff, N.R., Eccles, J., Lello, L., Bechor, E., Hsu, J., Plunkett, K., Zimmerman, R., Rana, B., Samoilenko, A., Hsu, S.*, et al.* (2019a). Utility and First Clinical Application of Screening Embryos for Polygenic Disease Risk Reduction. Front Endocrinol (Lausanne) *10*, 845.

Treff, N.R., Zimmerman, R., Bechor, E., Hsu, J., Rana, B., Jensen, J., Li, J., Samoilenko, A., Mowrey, W., Van Alstine, J.*, et al.* (2019b). Validation of concurrent preimplantation genetic testing for polygenic and monogenic disorders, structural rearrangements, and whole and segmental chromosome aneuploidy with a single universal platform. Eur J Med Genet.

Wang, Y., Guo, J., Ni, G., Yang, J., Visscher, P.M., and Yengo, L. (2020). Theoretical and empirical quantification of the accuracy of polygenic scores in ancestry divergent populations. Nat Commun *11*, 3865.

Wang, Y.F., Zhang, Y., Lin, Z., Zhang, H., Wang, T.Y., Cao, Y., Morris, D.L., Sheng, Y., Yin, X., Zhong, S.L.*, et al.* (2021). Identification of 38 novel loci for systemic lupus erythematosus and genetic heterogeneity between ancestral groups. Nat Commun *12*, 772.

Weissbrod, O., Kanai, M., Shi, H., Gazal, S., Peyrot, W., Khera, A., Okada, Y., Martin, A., Finucane, H., and Price, A.L. (2021). Leveraging fine-mapping and non-European training data to improve trans-ethnic polygenic risk scores. medRxiv, 2021.2001.2019.21249483.

Wojcik, G.L., Graff, M., Nishimura, K.K., Tao, R., Haessler, J., Gignoux, C.R., Highland, H.M., Patel, Y.M., Sorokin, E.P., Avery, C.L.*, et al.* (2019). Genetic analyses of diverse populations improves discovery for complex traits. Nature *570*, 514-518.

Xiong, L., Huang, L., Tian, F., Lu, S., and Xie, X.S. (2019). Bayesian model for accurate MARSALA (mutated allele revealed by sequencing with aneuploidy and linkage analyses). J Assist Reprod Genet *36*, 1263-1271.

Xu, Y., Chen, S., Yin, X., Shen, X., Pan, X., Chen, F., Jiang, H., Liang, Y., Wang, W., Xu, X.*, et al.* (2015). Embryo genome profiling by single-cell sequencing for preimplantation genetic diagnosis in a β-thalassemia family. Clin Chem *61*, 617-626.

Yan, L., Huang, L., Xu, L., Huang, J., Ma, F., Zhu, X., Tang, Y., Liu, M., Lian, Y., Liu, P.*, et al.* (2015). Live births after simultaneous avoidance of monogenic diseases and chromosome abnormality by next-generation sequencing with linkage analyses. Proc Natl Acad Sci U S A *112*, 15964-15969.

Zamani Esteki, M., Dimitriadou, E., Mateiu, L., Melotte, C., Van der Aa, N., Kumar, P., Das, R., Theunis, K., Cheng, J., Legius, E.*, et al.* (2015). Concurrent whole-genome haplotyping and copy-number profiling of single cells. American journal of human genetics *96*, 894-912.

Zhang, W.Y., von Versen-Hoynck, F., Kapphahn, K.I., Fleischmann, R.R., Zhao, Q., and Baker, V.L. (2019). Maternal and neonatal outcomes associated with trophectoderm biopsy. Fertil Steril *112*, 283-290 e282.

Zheng, J., Erzurumluoglu, A.M., Elsworth, B.L., Kemp, J.P., Howe, L., Haycock, P.C., Hemani, G., Tansey, K., Laurin, C., Early, G.*, et al.* (2017). LD Hub: a centralized database and web interface to perform LD score regression that maximizes the potential of summary level GWAS data for SNP heritability and genetic correlation analysis. Bioinformatics *33*, 272-279.